# FEDERATED LEARNING OF QUANTILE INFERENCE UNDER LOCAL DIFFERENTIAL PRIVACY

**Leheng Cai**[†]
Department of Statistics and Data Science
Tsinghua University
Beijing 100084, China
`cailh22@mails.tsinghua.edu.cn`

**Qirui Hu**[†*]
School of Statistics and Data Science,
Shanghai University of Finance and Economics
Institute of Big Data Research,
Shanghai University of Finance and Economics
Shanghai 200443, China
`huqirui@mail.shufe.edu.cn`

**Shuyuan Wu**[†]
School of Statistics and Data Science,
Shanghai University of Finance and Economics
Shanghai 200433, China
`wushuyuan@mail.sufe.edu.cn`

## ABSTRACT

In this paper, we investigate federated learning for quantile inference under local differential privacy (LDP). We propose an estimator based on local stochastic gradient descent (SGD), whose local gradients are perturbed via a randomized mechanism with global parameters, making the procedure tolerant of communication and storage constraints without compromising statistical efficiency. Although the quantile loss and its corresponding gradient do not satisfy standard smoothness conditions typically assumed in existing literature, we establish asymptotic normality for our estimator as well as a functional central limit theorem. The proposed method accommodates data heterogeneity and allows each server to operate with an individual privacy budget. Furthermore, we construct confidence intervals for the target value through a self-normalization approach, thereby circumventing the need to estimate additional nuisance parameters. Extensive numerical experiments and real data application validate the theoretical guarantees of the proposed methodology.

## 1 INTRODUCTION

Modern data ecosystems increasingly require distributional guarantees rather than simple averages. For instance, a national hospital network may track the 0.9 quantile of emergency waiting times across sites to ensure that "nine out of ten patients are seen within $T$ minutes" (Yadlowsky et al., 2025), while financial institutions assess tail risk via value-at-risk or expected shortfall under strict confidentiality constraints (Barbaglia et al., 2023; Chen, 2008; Wang et al., 2012). In both settings, the target is a quantile of a heterogeneous, possibly heavy-tailed distribution, and the scientific goals—comparing tail performance, checking compliance, or detecting post-intervention shifts—require full inferential tools, not just point estimates.

Quantile methods naturally support these tasks, revealing heterogeneity and tail behavior often invisible to means (Angrist et al., 2006; Kallus et al., 2024; Chernozhukov & Fernández-Val, 2011; Chernozhukov & Hansen, 2005; He et al., 2023; Hu et al., 2022; Chen et al., 2023). Yet the data required for such inference are increasingly distributed: hospitals, banks, and user-facing services each hold their own records, and centralizing raw data is often infeasible due to communication, storage, privacy, and regulatory barriers.

---

[*]Corresponding author
[†]The authors are listed in alphabetical order with equal contribution

These challenges motivate federated learning (McMahan et al., 2017; Konečnỳ et al., 2016; Liu et al., 2020; Tian et al., 2024), where a server coordinates updates from many clients without collecting their raw data. Local SGD can be optimal under i.i.d. data (Stich, 2019; Li et al., 2020; Chen et al., 2022), but realistic federated environments are heterogeneous in distributions, sample sizes, and even target quantiles, complicating optimization and inference and prompting methods based on regularization, momentum, and worst-case analysis (Hu et al., 2024; Li et al., 2020; 2025; 2022).

At the same time, protections limited to servers or silos are no longer considered sufficient. Breaches of medical and financial records show that server-side DP alone (Dwork et al., 2006) cannot prevent disclosure when the custodian is compromised. Local differential privacy (LDP) mitigates this risk by randomizing each individual record before transmission (Duchi et al., 2013); deployed systems such as Google's RAPPOR, Apple's telemetry, and Windows diagnostics demonstrate its practicality (Erlingsson et al., 2014; Ding et al., 2017).

Differentially private federated learning merges these ideas and has gained substantial interest (Agarwal et al., 2018; Shi et al., 2022; Ma et al., 2022; Liu et al., 2023a; 2024; Cai et al., 2025b). As emphasized by Lowy & Razaviyayn (2023), privacy guarantees vary by trust assumptions; LDP corresponds to the most conservative regime in which individuals trust neither server nor silo. Recent work studies LDP in distributed settings for generalized linear models, mean estimation, and related tasks (Zhao et al., 2021; Shen et al., 2023; Jiang et al., 2022), but primarily from an estimation perspective. For quantiles, existing LDP methods are either single-machine procedures that ignore client heterogeneity (Huang et al., 2021; Liu et al., 2023b) or provide point estimators without general inferential guarantees.

In contrast, our focus is on quantile inference under LDP in a federated, heterogeneous environment. Concretely, we consider a setting in which many clients, each holding local data drawn from potentially different distributions and targeting possibly different quantile levels, collaborate to estimate and infer a global quantile while ensuring that every message they send is locally private.

We illustrate with a simple example. Consider collaboratively estimating the national median annual income using state-level data from the United States, where each state is treated as a client. First, income distributions typically vary across states (see Figure 1(i)). Second, privacy preferences can differ across states due to cultural norms and development levels (Milberg et al., 2000; Bellman et al., 2004). Figure 1(ii) shows how the released information can vary under different privacy budgets. Due to such heterogeneity, a direct application of the divide-and-conquer method, which combines local LDP estimators from Liu et al. (2023b) designed for a single client, can yield estimation with significant bias and invalid inference.

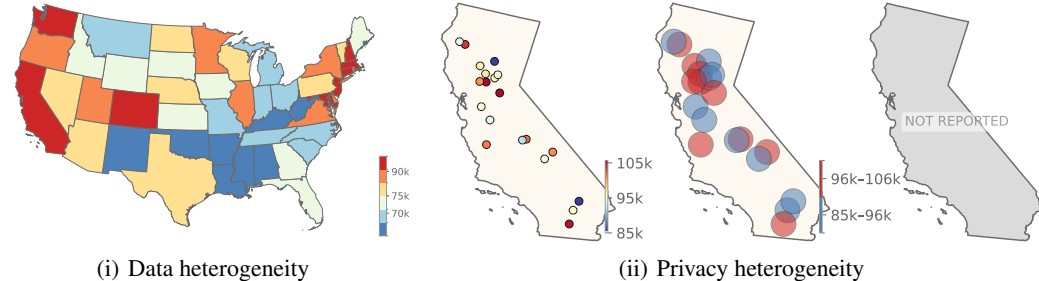

(i) Data heterogeneity                (ii) Privacy heterogeneity

Figure 1: Illustration of client heterogeneity. Income data source: U.S. Census Bureau (`https://data.census.gov/table/ACSST5Y2023.S1901?g=010XX00US0400000`). Panel (i) shows median annual income by state. Panel (ii) shows three income disclosure schemes under different privacy budgets: (a) each individual release true income; (b) each individual release an income interval; and (c) withholding release.

> The key question is: Can we design a federated procedure for quantile estimation under LDP that admits valid confidence intervals and hypothesis tests, while accommodating client heterogeneity and non-smooth loss functions?

Answering this question is challenging for at least three reasons. First, inference requires not only a limiting distribution for the estimator, but also a consistent estimator of its asymptotic variance. For SGD-based procedures, such variance estimators typically involve the Hessian of a smooth loss (Chen et al., 2020); yet quantile losses are non-smooth and heterogeneous across clients. Second, in the LDP regime, only privatized gradients are observed, so naive variance estimation may consume additional privacy budget or demand data-splitting. Third, federated quantile algorithms must remain robust to heterogeneous loss functions and client-level privacy parameters, which further complicates both optimization and asymptotic analysis.

In this paper, our contributions can be summarized as follows:

1. We propose a novel federated learning algorithm for quantile inference under LDP. Our method accommodates client-level heterogeneity in quantile targets, privacy budgets, and data distributions, thereby enhancing the applicability of quantile inference in realistic federated environments.

2. We first design an LDP mechanism that effectively reduces the federated quantile estimation problem to an equivalent non-private setting. Exploiting this reduction, we establish the estimator's asymptotic normality and derive a functional central limit theorem without average-smoothness condition on the loss function. To the best of our knowledge, this constitutes the first weak-convergence result for local SGD when the loss does not satisfy the usual average-smoothness condition (Li et al., 2022; Xie et al., 2024; Zhu et al., 2024).

3. Building on these non-private asymptotic results, we develop an LDP inference procedure for federated quantile estimation. By employing a self-normalization technique, we avoid direct estimation of the asymptotic variance, instead constructing confidence intervals that automatically eliminate the unknown variance term. These procedures can be also conducted in non-DP case.

The remainder of the paper is organized as follows. Section 2 introduces the formal problem setup, background, and notation, and presents our LDP mechanism and federated quantile estimator. Section 3 develops the asymptotic theory, including weak convergence and self-normalized inference. Section 4 reports numerical experiments and a real-data application illustrating the practical performance and robustness of our method. Notations can refer to Table S.1 in Appendix A.1. All technical proofs and additional simulation results are deferred to the Appendices B and C.

## 2 METHODOLOGIES

First, we recall the definitions of central and local differential privacy. We then describe our problem setting and algorithmic details.

**Definition 1** (Central Differential Privacy, CDP (Dwork et al., 2006)). *A randomized algorithm $\mathcal{A}$ operating on a dataset $S$ is $(\epsilon, \delta)$-differentially private if, for any pair of datasets $S$ and $S'$ differing in a single record and for any measurable set $E$,*

$$\Pr\big(\mathcal{A}(S) \in E\big) \leq e^\epsilon \Pr\big(\mathcal{A}(S') \in E\big) + \delta.$$

*When $\delta = 0$, $\mathcal{A}$ is called $\epsilon$-DP.*

**Definition 2** (Local Differential Privacy, LDP (Joseph et al., 2019)). *A family of randomized mappings $R : X \to Y$ is $(\epsilon, \delta)$-locally differentially private if, for every pair of inputs $x, x' \in X$ and every measurable subset $E \subseteq Y$,*

$$\Pr\big(R(x) \in E\big) \leq e^\epsilon \Pr\big(R(x') \in E\big) + \delta.$$

Under CDP, a trusted curator collects raw data and adds noise before release, simplifying algorithm design and typically incurring only an $\mathcal{O}(1/n)$ loss in accuracy (Cai et al., 2021), so the privatized estimator $\widehat{\theta}_{\mathrm{CDP}}$ often satisfies $\widehat{\theta}_{\mathrm{CDP}} - \widehat{\theta}_{\mathrm{nonDP}} = \mathcal{O}_p(n^{-1})$, shares the same $\sqrt{n}$-scaled asymptotic law as the non-private estimator, and permits recovery of its asymptotic variance at modest additional privacy cost. In contrast, under the non-adaptive LDP model (Cheu et al., 2019, Definitions 2.3 and 2.6), each user $i$ with private value $X_i$ applies a fixed randomized mechanism $R_i$ satisfying $(\epsilon, \delta)$-DP and reports only the perturbed output, so inference must rely solely on locally privatized

data; the typical error rate deteriorates to $\widehat{\theta}_{\text{LDP}} - \widehat{\theta}_{\text{nonDP}} = \mathcal{O}_p(n^{-1/2})$, which alters the limiting distribution, inflates the asymptotic variance, and generally makes consistent variance estimation from data collected solely for point estimation infeasible.

We consider a federated learning framework involving $K$ clients, each independently holding a local dataset i.i.d. drawn from an unknown distribution $\mathcal{P}_k$ with cumulative distribution function (CDF) $F_k$ and density function $f_k$ (Li et al., 2022). For $p_k$ weight assigned to client $k$, global quantile level $\tau \in (0, 1)$, the goal is to collaboratively estimate the global quantile $Q^\star$, which satisfies that $\sum_{k=1}^{K} p_k F_k(Q^\star) = \tau$. Our objective $Q^\star$ can be obtained by solving the following optimization problem associated with the global loss, which is defined as a weighted sum of the local losses:

$$Q^\star = \arg\min_{Q \in \Theta} \mathcal{L}(Q) \stackrel{\text{def}}{=} \arg\min_{Q \in \Theta} \sum_{k=1}^{K} p_k \mathcal{L}_{\tau_k}(Q) \stackrel{\text{def}}{=} \arg\min_{Q \in \Theta} \sum_{k=1}^{K} p_k \mathbb{E}_{x_k \sim \mathcal{P}_k}\{\ell_{\tau_k}(x_k, Q)\}. \quad (2.1)$$

Here, the function $\ell_{\tau_k}(x, Q)$ represents the check loss function defined as:

$$\ell_{\tau_k}(x, Q) = (x - Q)\{\tau_k - \mathbb{I}(x < Q)\}, \quad (2.2)$$

where $\mathbb{I}(\cdot)$ is the indicator function. Besides, $x_k$ is the sample generated from $\mathcal{P}_k$, and $\tau_k \in (0, 1)$ is the local quantile level satisfying $\sum_{k=1}^{K} p_k \tau_k = \tau$. Therefore, our framework also covers federated quantile loss optimization problems in which each local loss corresponds to a level $\tau_k$, and the global objective level $\tau$ aggregates these local target levels through $\sum_{k=1}^{K} p_k \tau_k$. Note that we require only the knowledge of $\tau$, rather than each individual $\tau_k$. In the following, we denote $F_k(Q^\star) = Q_k$, and consider the parameter space $\Theta$ is bounded; see Gu & Chen (2023).

As noted in the introduction, to improve communication efficiency we consider a local-SGD–based estimator. Let $\mathcal{I} = \{t_0, t_1, \ldots, t_T\}$ for $t_0 < t_1 < \cdots < t_T$ denote the ordered set of communication iterations. At each $t \in \mathcal{I}$, the global server receives the local updates and broadcasts the aggregated value to all $K$ clients; otherwise, the updates are performed locally on each client. Specifically, for $k = 1, \ldots, K$,

$$q_{t+1}^k = \begin{cases} q_t^k - \eta_t \{\mathbb{I}(x_t^k < q_t^k) - \tau_k\}, & t \notin \mathcal{I}, \\ \sum_{m=1}^{K} p_m [q_t^m - \eta_t \{\mathbb{I}(x_t^m < q_t^m) - \tau_m\}], & t \in \mathcal{I}. \end{cases}$$

Here $\eta_t$ is the predetermined learning rate, and $x_t^k$ is an observation from the local dataset held by client $k$ at iteration $t$. The final estimator is of Polyak–Ruppert type, obtained by averaging the aggregated historical iterates:

$$\widetilde{Q}_T = \frac{1}{T} \sum_{m=1}^{T} \sum_{k=1}^{K} p_k q_{t_m}^k.$$

The communication and statistical efficiency are determined by the interval length $E_m := t_m - t_{m-1}$ for $m \in \mathbb{N}^+$. If $E_m = 1$, the local clients must communicate with the global server at every iteration. In this scenario, the approach reduces to parallel SGD, which, as noted by Li et al. (2022), may achieve the Cramér-Rao lower bound and thus serve as an efficient estimator for certain smooth loss functions. Conversely, if communication is performed only once at the final iteration (i.e., $|\mathcal{I}| = 1$), then the estimator degenerates to a divide-and-conquer (DC) estimator. In this case, minimizing the loss function (2.1) becomes a distributed learning problem. However, as pointed out by Gu & Chen (2023), the divide-and-conquer estimator may still be statistically inefficient for certain weight choices. Therefore, a careful balance must be struck between communication and statistical efficiency. For a general positive interval $E_m > 0$, the local SGD method allows us to find an appropriate choice of $E_m$ that can ensure an optimal trade-off between these efficiencies.

On the other hand, the data collected from each client may be subject to privacy protection policies, particularly in surveys involving sensitive information such as income or health status. For the local quantile loss function (2.2), we observe that the structure of its gradient resembles a binary response. This motivates us to incorporate an LDP mechanism based on randomized response and permutation, following the framework of Liu et al. (2023b), with a truthful response rate $r_k \in (0, 1]$. Specifically, the mechanism allows each local client to either return a true gradient with probability $r_k$ or a synthetic Bernoulli random variable with probability $1 - r_k$. This iterative mechanism ensures

$\epsilon_k$-LDP, where the privacy parameter is given by $\epsilon_k = \log(1+r_k) - \log(1-r_k)$, as established in Liu et al. (2023b). Combined with local SGD, the complete procedure is summarized in Algorithm 1, and we denote the resulting estimator as $\widehat{Q}_T$. By direct composition properties, Algorithm enjoys $(\max_{1 \le k \le K} \epsilon_k, 0)$-LDP guarantees.

---

Algorithm 1: Federated quantile estimation with local SGD under LDP

---

**Input:** step sizes $\{\eta_m\}_{m=0}^T$, target quantile $\tau \in (0,1)$, truthful response rates $\{r_k\}_{k=1}^K$, communication set $\mathcal{I} = \{t_0, t_1, \ldots, t_T\}$.

**Initialization:** set $q_0^k = q_0 \sim \mathcal{N}(0,1)$ for all $1 \le k \le K$, let $\widehat{Q}_0 \leftarrow 0$.

  **for** $m = 1$ to $T$ **do**

    **for** $k = 1$ to $K$ (distributedly) **do**

      **for** $t = t_{m-1} + 1$ to $t_m$ **do**                         ▷ Local updates

        $u_t^k \sim Bernoulli(r_k), \quad v_t^k \sim Bernoulli(0.5)$

        $s_t^k = \mathbb{I}(x_t^k > q_{t-1}^k)\mathbb{I}(u_t^k = 1) + v_t^k \mathbb{I}(u_t^k = 0)$

        $q_t^k \leftarrow q_{t-1}^k + \dfrac{1 - r_k + 2\tau r_k}{2r_k}\eta_{m-1}\mathbb{I}(s_t^k = 1) - \dfrac{1 + r_k - 2\tau r_k}{2r_k}\eta_{m-1}\mathbb{I}(s_t^k = 0)$

      **end for**

    **end for**

    $\bar{q}_{t_m} \leftarrow \sum_{k=1}^K p_k q_{t_m}^k ; q_{t_m}^k \leftarrow \bar{q}_{t_m}$ for all $1 \le k \le K$.     ▷ Aggregation and synchronization.

    $\widehat{Q}_m \leftarrow \{(m-1)\widehat{Q}_{m-1} + \bar{q}_{t_m}\}/m$.

  **end for**

**Return:** $\widehat{Q}_T$.

---

In Algorithm 1, each iteration integrates global information (the global quantile $\tau$) with client-specific data $x_t^k$ and corresponding privacy budget ($r_k$), thereby correcting bias arising from the aggregation of heterogeneous local LDP mechanisms and loss functions.

**Theorem 2.1.** *Denote $\widetilde{\tau}_k = r_k \tau + (1 - r_k)/2$. For a privacy budget $\epsilon_k = \log(1+r_k) - \log(1-r_k)$, there exists a dataset consisting of i.i.d. samples drawn from some distribution $\widetilde{\mathcal{P}}_k, 1 \le k \le K$, such that solving the federated loss (2.1) with $\epsilon_k$-LDP using data drawn from $\mathcal{P}_k$ for client $k$, is equivalent to solving the following non-private problem:*

$$Q^\star = \arg\min_Q \mathcal{L}(Q) = \arg\min_Q \widetilde{\mathcal{L}}(Q) \stackrel{\text{def}}{=} \arg\min_Q \sum_{k=1}^K \frac{p_k}{r_k} \mathbb{E}_{x_k \sim \widetilde{\mathcal{P}}_k} \{\ell_{\widetilde{\tau}_k}(x_k, Q)\}. \tag{2.3}$$

By Theorem 2.1, federated quantile estimation via (2.1) with LDP data drawn from $\{\mathcal{P}_k\}_{k=1}^K$ can be reformulated as federated quantile estimation with non-DP data drawn from modified distributions $\{\widetilde{\mathcal{P}}_k\}_{k=1}^K$ and shifted quantile levels, as in (2.3). Consequently, the target value $Q^\star$ can be treated as the minimizer of $\widetilde{\mathcal{L}}(Q)$, a non-DP objective. The main challenge reduces to analyzing the statistical properties of the resulting non-DP estimator (under the ideal data from $\{\widetilde{\mathcal{P}}_k\}_{k=1}^K$), particularly in the presence of the non-smooth quantile loss function. Once such properties are established, the LDP inference theory can be translated directly from the relation between $(\{\mathcal{P}_k\}_{k=1}^K, \tau)$ and $(\{\widetilde{\mathcal{P}}_k\}_{k=1}^K, \{\widetilde{\tau}_k\}_{k=1}^K)$.

## 3   Asymptotic analysis

In this section, we focus on the asymptotic analysis of the proposed LDP estimator and the practical construction of confidence intervals. Due to the space limit, the complete assumptions and corresponding comments are provided in Appendix A.2.

**Theorem 3.1.** *Under Assumptions S.1-S.3, as $T \to \infty$, the proposed LDP federated estimator enjoys*

$$\sqrt{t_T}(\widehat{Q}_T - Q^\star) \xrightarrow{d} N\left(0, \nu \frac{\sum_{k=1}^K p_k^2 \{r_k^{-2} - (2Q_k - 1)^2\}}{4\left\{\sum_{k=1}^K p_k f_k(Q^\star)\right\}^2}\right).$$

Theorem 3.1 establishes the asymptotic normality of the estimator $\widehat{Q}_T$. It follows that the convergence rate of $\widehat{Q}_T$ is of order $(\min_{1 \leq k \leq K} r_k\, t_T)^{-1/2}$. Recall that the proposed algorithm satisfies $(\max_{1 \leq k \leq K} \log\{(1+r_k)/(1-r_k)\}, 0)$-LDP. which implies that the overall rate is determined by the client with the largest privacy budget. Smaller values of $r_k$ correspond to stronger privacy protection but inevitably yield poorer estimation accuracy. If there exists any $r_k = 0$, the variance diverges and the estimator becomes inconsistent. In contrast, if all $r_k = 1$, the result recovers the classical non-DP asymptotic normality. This highlights the fundamental privacy–accuracy trade-off inherent in the proposed estimator.

Theorem 3.1 allows for the theoretical construction of a confidence interval for $Q^\star$. However, the construction involves unknown quantities, such as the individual quantiles $Q_k$ and the density values $f_k(Q^\star)$. Even in cases where $Q_k$ is known, the estimation of $f_k(Q^\star)$ remains challenging. In particular, it is difficult to recover these density values using only the perturbed gradients available from Algorithm 1. Moreover, in SGD-based methods, consistent variance estimation typically relies on the Hessian matrix, which is well-defined only for smooth loss functions, as previously discussed. Therefore, although Theorem 3.1 provides a theoretically valid basis for confidence interval construction, it is not practically implementable due to these limitations.

To address this issue, we establish a functional central limit theorem (FCLT) in Theorem 3.2, which serves as the foundation for constructing pivotal statistics via self-normalization (Liu et al., 2023b). This allows us to conduct inference without estimating nuisance parameters, thereby providing a direct motivation for strengthening the asymptotic result.

**Theorem 3.2.** *Under Assumptions S.1–S.3, as $T \to \infty$, we have*

$$\mathcal{Q}_T(s) := \frac{\sqrt{t_T}}{T} \sum_{m=1}^{h(s,T)} (\bar{q}_{t_m} - Q^\star) \xrightarrow{d} \frac{\sqrt{\nu \sum_{k=1}^{K} p_k^2 \left\{ r_k^{-2} - (2Q_k - 1)^2 \right\}}}{2 \sum_{k=1}^{K} p_k f_k(Q^\star)} B(s),$$

*where $t_T = \sum_{m=0}^{T-1} E_m$, $\bar{q}_{t_m} = \sum_{k=1}^{K} p_k q_{t_m}^k$, $B(\cdot)$ is a standard Brownian motion on $[0,1]$, and*

$$h(s,T) = \max \left\{ n \in \mathbb{Z}_{>0} \,\middle|\, s \sum_{m=1}^{T} \frac{1}{E_m} \geq \sum_{m=1}^{n} \frac{1}{E_m} \right\}, \quad \text{for } s \in (0,1].$$

Theorem 3.2 establishes a FCLT for $Q_T(s)$ over $s \in (0,1]$, showing that it converges weakly in the $\ell^\infty[0,1]$ (the space of bounded real-valued functions) to a Brownian motion, which is our another major theoretical contribution. Note that the sample quantile loss doesn't satisfy the common $L$-average smooth conditions for weakly convergence result, such in (Li et al., 2022; Xie et al., 2024; Zhu et al., 2024), leading to extra challenge in deriving the almost sure and $\mathcal{L}^2$ convergence rates of $\bar{q}_{t_m}$, which are essential for handling the asymptotically negligible terms. Theorem 3.1 arises as a special case of Theorem 3.2 when $s = 1$. Building on Theorem 3.2, we proceed to construct a self-normalized test statistic and derive its asymptotic pivotal distribution via the continuous mapping theorem.

Define $r_0 = 0$ and, for $m \geq 1$, $r_m = \left( \sum_{i=1}^{m} 1/E_i \right) \left( \sum_{i=1}^{T} 1/E_i \right)^{-1}$, which ensures that

$$\mathcal{Q}_T(r_m) = \frac{\sqrt{t_T}}{T} \sum_{i=1}^{m} (\bar{q}_{t_i} - Q^\star), \quad \text{and in particular,} \quad \mathcal{Q}_T(1) = \frac{\sqrt{t_T}}{T} \sum_{i=1}^{T} (\bar{q}_{t_i} - Q^\star).$$

Following the arguments in (Shao, 2015), once a functional central limit theorem such as Theorem 3.2 is established, one can construct a self-normalized statistic that asymptotically enjoys a pivotal distribution. Specifically, define

$$\mathcal{V}_T = \sum_{m=1}^{T} (r_m - r_{m-1}) \left\{ \mathcal{Q}_T(r_m) - \frac{m}{T} \mathcal{Q}_T(1) \right\}^2. \tag{3.1}$$

**Corollary 3.1.** *Suppose Assumptions S.1-S.3 hold and $g(r_m) \asymp m/T$ for some continuous function $g$ on $[0,1]$. Then, as $T \to \infty$,*

$$\frac{\mathcal{Q}_T(1)}{\sqrt{\mathcal{V}_T}} \xrightarrow{d} \frac{B(1)}{\sqrt{\int_0^1 \left\{ B(r) - g(r) B(1) \right\}^2 dr}}.$$

Corollary 3.1 presents the asymptotic distribution of the self-normalized statistic $Q_T(1)/\mathcal{V}_T$, which is distribution-free. As a result, there is no need to allocate additional DP budget to estimate nuisance parameters when constructing confidence intervals.

The selection of the self-normalizer is not unique, and an appropriate norm of the Gaussian process $B(r) - g(r)B(1)$ can yield similar results to those in Corollary 3.1. For example, using the supremum norm and the $\mathcal{L}_1$ norm, one can define alternative self-normalizers as follows:

$$\mathcal{V}_T' = \sup_{1 \leq m \leq T} \left| \mathcal{Q}_T(r_m) - \frac{m}{T}\mathcal{Q}_T(1) \right|, \quad \mathcal{V}_T'' = \sum_{m=1}^{T} (r_m - r_{m-1}) \left| \mathcal{Q}_T(r_m) - \frac{m}{T}\mathcal{Q}_T(1) \right|,$$

which are related to the processes $\sup_{0 \leq r \leq 1} |B(r) - g(r)B(1)|$ and $\int_0^1 |B(r) - g(r)B(1)|\, dr$, respectively. However, the self-normalizer defined in equation (3.1) enjoys greater computational efficiency, as the $\mathcal{L}_2$ norm can be computed in an online manner, as described in Algorithm 2. Let $\widehat{\mathcal{V}}_T$ denote the estimator of the self-normalizer in (3.1), and let $v_{\alpha/2,g}$ be the $(1 - \alpha/2)$ quantile of the random variable $B(1)/ \left[ \int_0^1 \{B(r) - g(r)B_1(1)\}^2\, dr \right]^{1/2}$. The following corollary ensures the asymptotic validity of the constructed LDP confidence interval.

The following corollary ensures the asymptotic validity of the constructed LDP confidence interval.

**Corollary 3.2.** *Suppose the same conditions in Theorem 3.2 hold, as $T \to \infty$, one has that*

$$\mathbb{P}\left( \widehat{Q}_T - v_{\frac{\alpha}{2},g}\sqrt{\widehat{\mathcal{V}}_T} \leq Q^\star \leq \widehat{Q}_T + v_{\frac{\alpha}{2},g}\sqrt{\widehat{\mathcal{V}}_T} \right) \to 1 - \alpha$$

---

### Algorithm 2: Online Inference

---

**Input:** step sizes $\{\eta_m\}_{m=1}^T$, target quantile $\tau \in (0,1)$, truthful response rates $\{r_k\}_{k=1}^K$, communication set $\mathcal{I} = \{t_0, t_1, \ldots, t_T\}$.
**Initialization:** set $q_0^k \sim \mathcal{N}(0,1)$ for all $k$, let $\mathcal{V}_0^a \leftarrow 0$, $\mathcal{V}_0^b \leftarrow 0$, $\mathcal{V}_0^s \leftarrow 0$, $\mathcal{V}_0^p \leftarrow 0$, and $Q_0 \leftarrow 0$.
   **for** $m = 1$ to $T$ **do**
      Obtain $\widehat{Q}_m$ from Algorithm 1.
      $\mathcal{V}_m^a \leftarrow \mathcal{V}_{m-1}^a + m^2 Q_m^2/E_m, \qquad \mathcal{V}_m^b \leftarrow \mathcal{V}_{m-1}^b + m^2 Q_m/E_m, \qquad\qquad \triangleright\, E_m = t_m - t_{m-1}$
      $\mathcal{V}_m^s \leftarrow \mathcal{V}_{m-1}^s + 1/E_m \qquad \mathcal{V}_m^p \leftarrow \mathcal{V}_{m-1}^p + m^2/E_m.$
      $\widehat{\mathcal{V}}_m \leftarrow \frac{1}{m^2 \mathcal{V}_m^s}\left( \mathcal{V}_m^a - 2\mathcal{V}_m^b Q_m + \mathcal{V}_m^p Q_m^2 \right).$            $\triangleright$ Online inference.
   **end for**
**Return:** Confidence interval $\left[ \widehat{Q}_T - v_{\frac{\alpha}{2},g}\sqrt{\widehat{\mathcal{V}}_T}, \;\; \widehat{Q}_T + v_{\frac{\alpha}{2},g}\sqrt{\widehat{\mathcal{V}}_T} \right].$

---

## 4 EXPERIMENTS

### 4.1 SIMULATION SETUP

We first evaluate our proposed method through extensive simulation studies using synthetic data. In all experiments, we fixed $p_k = 1/K$ for $1 \leq k \leq K$, the number of clients is fixed at $K = 10$. The quantile levels examined range from 0.3 to 0.8, and the truthful response rates vary between 0.25 and 0.9. We focus on the following four scenarios of heterogeneity:

- **heterogeneous quantile levels:** We investigate two distinct scenarios: (1) Case $\tau_{\text{low}}$: lower quantile levels, where each client is assigned a unique quantile level $\tau_k$ ranging uniformly from 0.3 to 0.5; and (2) Case $\tau_{\text{high}}$: higher quantile levels, where $\tau_k$ ranges uniformly from 0.5 to 0.8.

- **heterogeneous response rates.** Each client has a unique truthful response rate $r_k$, ranging uniformly from 0.25 to 0.9.

- **heterogeneous locations (Hete L).** Data for each client $k$ are independently generated from $\mathcal{N}(\mu_k, 1)$ or $\mathcal{C}(\mu_k, 1)$, where $\mu_k \sim \mathcal{N}(0, 1)$.

- **heterogeneous distribution families (Hete D).** Data are generated independently across ten clients, with three drawing from $\mathcal{N}(0,1)$, three from the uniform distribution $\mathcal{U}(-1,1)$, and four from a standard Cauchy distribution $\mathcal{C}(0,1)$.

We set the step size $\gamma_m$ as: $\gamma_m = 20\bar{r}/(m^{0.51} + 100)$, with $\gamma_m = E_m\eta_m$ and $\bar{r} = K^{-1}\sum_{k=1}^{K} r_k$. Following Li et al. (2022), we implement a warm-up phase, setting the communication interval $E_m = 1$ for the first 5% of iterations. After the warm-up period, we redefine the interval sequence $\{E_m\}$ based on a new sequence $\{E'_m\}$, specifically: $E_m = E'_{m-0.05\cdot T}$. We examine three different interval strategies for $E'_m$: (1) C1: $E'_m \equiv 1$ (equivalent to parallel SGD), (2) C5: $E'_m \equiv 5$, and (3) Log: $E'_m = \lceil \log_2(m+1) \rceil$. The initial parameter estimates are set to $q_0^k = q_0 \sim \mathcal{N}(0,1)$ for all clients $k$. All experimental settings are replicated $R = 1,000$ times. The simulations are conducted on computational resources comprising 36 Intel Xeon Gold 6271 CPUs, with a total of 128GB RAM and 500GB storage.

## 4.2 Simulation Results

We first illustrate the performance of our proposed method by presenting sample iteration trajectories for estimation and inference. Specifically, we randomly select one simulation run and plot the resulting estimates and corresponding confidence intervals against $t_T$ (Figure S.1). The results demonstrate that our approach accurately captures the true quantile value and provides reliable inference. Subsequently, we fix $t_T$ at $10,000$ and $50,000$ and evaluate the finite sample performance under different settings. Let $\widehat{Q}_T^{(r)}$ denote the quantile estimator and $\text{CI}^{(r)}$ represent the corresponding 95% confidence interval obtained from Algorithm 2 in the $r$-th simulation. We consider two metrics: the mean absolute error (MAE), defined as $R^{-1}\sum_{r=1}^{R}|\widehat{Q}_T^{(r)} - Q^*|$, and the empirical coverage probability (ECP), defined as $R^{-1}\sum_{r=1}^{R}\mathbb{I}(Q^* \in \text{CI}^{(r)})$. For comparison, we also consider two alternative methods: (1) the SGD with DP updates (Song et al., 2013) (DP-SGD), which adds noise directly to the gradients instead of introducing DP through randomized response. To align with the original paper's setup, we focus on the case with $C = 1$. In this regime, the gradient-descent update in Algorithm 1 becomes

$$q_t^k \leftarrow q_{t-1}^k + \eta_{t-1}\Big\{\tau_k\mathbb{I}(x_t^k > q_{t-1}^k) - (1-\tau_k)\mathbb{I}(x_t^k < q_{t-1}^k) + Z_t^k\Big\},$$

where $Z_t^k$ is drawn from a Laplace distribution. A simple calculation shows that $Z_t^k$ has mean zero and scale parameter $1/\log\{(1+r_k)/(1-r_k)\}$. (2) the divide-and-conquer (DC) method, which corresponds to the special case $E_m = t_T$. Here we use step size $\eta_t = 2\bar{r}/(t^{0.51} + 100)$ (Goyal et al., 2017). (3) the single-machine LDP quantile inference method (Single) (Liu et al., 2023b), where all data hold on a single device and LDP-SGD is performed without any federated communication. The numerical results for all methods under the normal distribution setting are reported in Tables 1 and 2, while the corresponding results under the Cauchy distribution setting are summarized in Tables S.2 and S.3.

From the results, we observe that our method consistently achieves ECP close to or exceeding the nominal 95% level across all scenarios, and that the C1 strategy (parallel SGD) performs essentially comparably to the Single baseline. As either the total sample size $t_T$ or the truthful response rate increases, the MAE decreases, which aligns with our theoretical results. Comparing the three interval strategies, we find that the C1 strategy yields the smallest MAE, as it has the highest communication frequency. Comparing with the two competing methods, we find that the DC approach results in the largest errors. Notably, in certain heterogeneous cases, such as Hete L with $\tau = 0.8$, the DC estimator exhibits significant bias and an ECP far below the nominal 95% level. In contrast, our proposed estimators successfully achieve approximately 95% empirical coverage in these cases. Moreover, while DP-SGD attains empirical coverage probabilities close to or even exceeding 95% in most settings, its MAE remain uniformly larger than those of our method.

To further illustrate the efficiency–accuracy trade-off of our method, we conduct a quantitative analysis in which we record the wall-clock time; the results are shown in Figure S.2 in Appendix C.1. We also consider scenarios with a fixed number of communication rounds T, with the corresponding results summarized in Tables S.4 and S.5 in Appendix C.1. We observe that our proposed method continues to provide valid inference. Additionally, under fixed communication rounds, the Log strategy generally achieves the best performance, yielding the smallest MAE.

| Quantile ($\tau$) | Rate ($r$) | C1 | C5 | Log | DP-SGD (C1) | DC | Single |
|---|---|---|---|---|---|---|---|
| | | | | Hete L — $t_T = 10000$ | | | |
| 0.3 | 0.25 | 0.958(0.0184) | 0.981(0.0311) | 0.990(0.0452) | 0.942(0.0260) | 0.985(0.3066) | 0.950(0.0222) |
| 0.3 | hetero | 0.949(0.0096) | 0.982(0.0150) | 0.993(0.0205) | 0.947(0.0142) | 0.898(0.1302) | 0.954(0.0095) |
| 0.3 | 0.9 | 1.000(0.0029) | 1.000(0.0066) | 1.000(0.0100) | 0.981(0.0049) | 0.215(0.1273) | 0.962(0.0055) |
| 0.5 | 0.25 | 0.950(0.0165) | 0.984(0.0315) | 0.988(0.0465) | 0.953(0.0224) | 1.000(0.2822) | 0.930(0.0192) |
| 0.5 | hetero | 0.952(0.0085) | 0.991(0.0155) | 0.998(0.0221) | 0.955(0.0119) | 1.000(0.0525) | 0.952(0.0083) |
| 0.5 | 0.9 | 0.996(0.0025) | 0.999(0.0078) | 1.000(0.0120) | 0.984(0.0041) | 1.000(0.0186) | 0.952(0.0051) |
| 0.8 | 0.25 | 0.966(0.0237) | 0.995(0.0512) | 0.992(0.0791) | 0.957(0.0328) | 0.892(0.6152) | 0.942(0.0277) |
| 0.8 | hetero | 0.962(0.0122) | 0.995(0.0227) | 0.996(0.0347) | 0.943(0.0186) | 0.709(0.2684) | 0.958(0.0114) |
| 0.8 | 0.9 | 0.990(0.0042) | 1.000(0.0116) | 1.000(0.0185) | 0.968(0.0065) | 0.049(0.2098) | 0.966(0.0067) |
| | | | | Hete L — $t_T = 50000$ | | | |
| 0.3 | 0.25 | 0.937(0.0089) | 0.981(0.0111) | 0.990(0.0165) | 0.916(0.0135) | 0.949(0.1328) | 0.948(0.0095) |
| 0.3 | hetero | 0.911(0.0056) | 0.981(0.0056) | 0.997(0.0080) | 0.885(0.0083) | 0.093(0.1282) | 0.958(0.0038) |
| 0.3 | 0.9 | 0.977(0.0034) | 1.000(0.0019) | 1.000(0.0030) | 0.908(0.0041) | 0.000(0.1290) | 0.938(0.0024) |
| 0.5 | 0.25 | 0.958(0.0069) | 0.988(0.0098) | 0.995(0.0147) | 0.949(0.0099) | 1.000(0.0609) | 0.940(0.0082) |
| 0.5 | hetero | 0.964(0.0035) | 0.994(0.0048) | 0.996(0.0069) | 0.957(0.0052) | 0.997(0.0145) | 0.938(0.0034) |
| 0.5 | 0.9 | 1.000(0.0010) | 1.000(0.0016) | 1.000(0.0026) | 0.993(0.0018) | 0.979(0.0143) | 0.950(0.0022) |
| 0.8 | 0.25 | 0.956(0.0102) | 0.991(0.0144) | 0.998(0.0226) | 0.931(0.0160) | 0.799(0.2829) | 0.954(0.0114) |
| 0.8 | hetero | 0.950(0.0055) | 0.992(0.0072) | 0.997(0.0112) | 0.923(0.0092) | 0.014(0.2034) | 0.952(0.0046) |
| 0.8 | 0.9 | 1.000(0.0013) | 1.000(0.0053) | 0.999(0.0082) | 0.985(0.0026) | 0.000(0.1929) | 0.950(0.0027) |
| | | | | Hete D — $t_T = 10000$ | | | |
| 0.5 | 0.25 | 0.949(0.0132) | 0.985(0.0243) | 0.986(0.0354) | 0.953(0.0183) | 0.904(0.2496) | 0.936(0.0139) |
| 0.5 | hetero | 0.966(0.0069) | 0.990(0.0117) | 0.989(0.0172) | 0.955(0.0098) | 0.999(0.0488) | 0.958(0.0057) |
| 0.5 | 0.9 | 1.000(0.0023) | 1.000(0.0074) | 1.000(0.0117) | 0.991(0.0035) | 1.000(0.0163) | 0.946(0.0037) |
| | | | | Hete D — $t_T = 50000$ | | | |
| 0.5 | 0.25 | 0.958(0.0057) | 0.980(0.0082) | 0.993(0.0127) | 0.943(0.0081) | 0.981(0.0589) | 0.940(0.0058) |
| 0.5 | hetero | 0.966(0.0030) | 0.988(0.0046) | 0.998(0.0073) | 0.950(0.0041) | 1.000(0.0111) | 0.958(0.0025) |
| 0.5 | 0.9 | 0.999(0.0008) | 1.000(0.0029) | 1.000(0.0052) | 0.990(0.0014) | 1.000(0.0037) | 0.950(0.0016) |

Table 1: Empirical coverage probabilities at the 95% nominal level (mean absolute errors ↓) under heterogeneous distributions for different $t_T$. The number of clients $K$ is fixed at 10. In Hete L, data for each client $k$ are independently generated from $\mathcal{N}(\mu_k, 1)$, where $\mu_k \sim \mathcal{N}(0, 1)$. In Hete D, data are generated from $\mathcal{N}(0, 1)$, $\mathcal{U}(-1, 1)$, and $\mathcal{C}(0, 1)$ across different clients. "hetero" indicates client-specific truthful response rates $r_k$ range uniformly from $[0.25, 0.9]$.

| Quantile ($\tau$) | Rate ($r$) | C1 | C5 | Log | DP-SGD (C1) | DC | Single |
|---|---|---|---|---|---|---|---|
| | | | | $t_T = 10000$ | | | |
| 0.5 | 0.25 | 0.949(0.0133) | 0.967(0.0244) | 0.992(0.0360) | 0.949(0.0191) | 0.939(0.2503) | 0.934(0.0141) |
| 0.5 | hetero | 0.963(0.0071) | 0.989(0.0112) | 0.997(0.0161) | 0.955(0.0100) | 1.000(0.0497) | 0.956(0.0059) |
| 0.5 | 0.9 | 0.995(0.0023) | 1.000(0.0054) | 1.000(0.0082) | 0.980(0.0036) | 1.000(0.0158) | 0.950(0.0039) |
| $\tau_{\text{low}}$ | 0.25 | 0.947(0.0136) | 0.982(0.0253) | 0.990(0.0369) | 0.940(0.0200) | 0.969(0.2616) | 0.950(0.0141) |
| $\tau_{\text{low}}$ | hetero | 0.962(0.0072) | 0.993(0.0113) | 0.997(0.0162) | 0.949(0.0105) | 0.999(0.0530) | 0.946(0.0063) |
| $\tau_{\text{low}}$ | 0.9 | 0.999(0.0020) | 1.000(0.0055) | 1.000(0.0083) | 0.985(0.0036) | 1.000(0.0162) | 0.942(0.0040) |
| $\tau_{\text{high}}$ | 0.25 | 0.939(0.0145) | 0.987(0.0268) | 0.986(0.0399) | 0.952(0.0210) | 0.984(0.2771) | 0.930(0.0152) |
| $\tau_{\text{high}}$ | hetero | 0.968(0.0076) | 0.987(0.0126) | 0.999(0.0182) | 0.956(0.0111) | 1.000(0.0516) | 0.972(0.0061) |
| $\tau_{\text{high}}$ | 0.9 | 0.996(0.0023) | 1.000(0.0067) | 1.000(0.0102) | 0.980(0.0038) | 1.000(0.0172) | 0.962(0.0039) |
| | | | | $t_T = 50000$ | | | |
| 0.5 | 0.25 | 0.956(0.0056) | 0.982(0.0081) | 0.996(0.0122) | 0.949(0.0081) | 0.988(0.0571) | 0.952(0.0058) |
| 0.5 | hetero | 0.960(0.0032) | 0.979(0.0044) | 0.992(0.0064) | 0.950(0.0046) | 1.000(0.0115) | 0.960(0.0025) |
| 0.5 | 0.9 | 1.000(0.0018) | 0.988(0.0027) | 0.990(0.0036) | 0.983(0.0021) | 1.000(0.0038) | 0.950(0.0016) |
| $\tau_{\text{low}}$ | 0.25 | 0.957(0.0061) | 0.981(0.0083) | 0.994(0.0125) | 0.944(0.0091) | 0.993(0.0594) | 0.944(0.0059) |
| $\tau_{\text{low}}$ | hetero | 0.953(0.0036) | 0.981(0.0054) | 0.990(0.0066) | 0.934(0.0053) | 0.999(0.0121) | 0.962(0.0026) |
| $\tau_{\text{low}}$ | 0.9 | 1.000(0.0019) | 1.000(0.0026) | 0.989(0.0038) | 0.988(0.0024) | 1.000(0.0057) | 0.958(0.0017) |
| $\tau_{\text{high}}$ | 0.25 | 0.968(0.0059) | 0.986(0.0086) | 0.997(0.0133) | 0.946(0.0089) | 0.999(0.0620) | 0.944(0.0061) |
| $\tau_{\text{high}}$ | hetero | 0.953(0.0032) | 0.990(0.0045) | 0.998(0.0065) | 0.952(0.0047) | 0.993(0.0154) | 0.948(0.0028) |
| $\tau_{\text{high}}$ | 0.9 | 0.998(0.0010) | 0.999(0.0023) | 1.000(0.0034) | 0.977(0.0016) | 0.938(0.0132) | 0.958(0.0017) |

Table 2: Empirical coverage probabilities at the 95% nominal level (mean absolute errors ↓) under varying quantile levels and response rates, with different $t_T$ and fixed $K = 10$ clients and data generated from $\mathcal{N}(0, 1)$. In Case $\tau_{\text{low}}$, each client uses a unique quantile level $\tau_k$ ranging uniformly from $[0.3, 0.5]$; in Case $\tau_{\text{high}}$, $\tau_k$ is ranging from $[0.5, 0.8]$. "hetero" indicates client-specific truthful response rates $r_k$ range uniformly from $[0.25, 0.9]$.

Finally, to further strengthen our simulation study, we conducted a set of additional experiments, including: (1) consider partial client participation to mimic more realistic federated environments; (2) sensitivity analyses on the truthful response rate and step-size schedule to assess the robustness of the method; (3) a comparison between the resulting confidence intervals and oracle normal-based intervals to illustrate the conservativeness of the self-normalized inference procedure; and (4) an initial exploration of extending our method to a fully decentralized federated learning setting; see Appendix C for details.

## 4.3 REAL DATA

In this subsection, we empirically evaluate the effectiveness of our proposed method using a representative real-world dataset widely employed in privacy research: Government Salary Dataset (Plečko et al., 2024). This dataset is sourced from the 2018 American Community Survey conducted by the U.S. Census Bureau and contains over 200,000 records, with annual salary (in USD) as the response variable. Since annual salary represents sensitive financial information (Gillenwater et al., 2021), we treat it as requiring privacy protection. To incorporate the dataset's inherent geographic structure, we partition the sample according to the feature "economic region." The three smallest regions are merged into a single "Others" category, yielding seven regions in total, each regarded as one client. Because region-level sample sizes vary, we apply oversampling to balance the data, resulting in $t_T = 53,960$ observations per client. All other hyperparameters follow the settings in Section 4.1. For analysis, we apply a log transformation to the response variable and subsequently back-transform it.

We target quantile levels $\tau_k \equiv \tau \in \{0.3, 0.5, 0.8\}$ and consider response rates $r_k \in \{0.6, \text{hetero}, 0.9\}$, where "hetero" indicates client-specific truthful response rates $r_k$ range uniformly from 0.6 to 0.9. For reference, we also compute the full-sample quantiles without LDP. The resulting estimators and confidence-interval lengths are summarized in Table 3. As shown, higher response rates $r$ and more communication rounds generally produce shorter confidence intervals, consistent with our simulation findings. In most cases, the empirical quantiles fall within our reported intervals, highlighting the practical utility of our method for real data.

| Quantile ($\tau$) | Rate ($r$) | C1 | C5 | Log | Empirical |
|---|---|---|---|---|---|
| 0.3 | 0.6 | 33367 (1742) | 33184 (6697) | 33030 (12093) | |
| 0.3 | hetero | 33418 (1424) | 33229 (5291) | 33140 (9788) | 34000 |
| 0.3 | 0.9 | 33547 (1548) | 33403 (4443) | 33239 (7828) | |
| 0.5 | 0.6 | 48454 (2255) | 48212 (6315) | 47951 (11361) | |
| 0.5 | hetero | 48462 (1435) | 48290 (4973) | 48091 (9025) | 50000 |
| 0.5 | 0.9 | 48610 (1454) | 48494 (3851) | 48311 (6863) | |
| 0.8 | 0.6 | 78586 (2066) | 78168 (6646) | 77995 (13144) | |
| 0.8 | hetero | 78390 (1291) | 78054 (5862) | 77722 (11101) | 80000 |
| 0.8 | 0.9 | 78657 (1138) | 78300 (4677) | 78084 (8928) | |

Table 3: Estimation results (interval lengths) on the real dataset across varying quantile levels and response rates. "Empirical" denotes the full-sample quantile estimator without LDP. "hetero" indicates client-specific truthful response rates $r_k$ range uniformly from 0.6 to 0.9.

## 5 CONCLUDING REMARK

We propose a federated-learning algorithm for quantile inference under LDP that flexibly accommodates client-level heterogeneity in quantile targets, privacy budgets, and data distributions. In addition, one innovation that should be emphasized is that our developed theoretical results of local SGD quantile estimator. We first design an LDP mechanism that can transform the LDP federated quantile estimation into the non-DP case, and then derive the asymptotic normality and functional central limit theorem of the proposed estimator under non-DP cases. It is first weak-convergence result for local SGD without the usual average-smoothness assumption in existing literature. Building on these non-private asymptotic results, we develop a self-normalized inference procedure that constructs valid confidence intervals under LDP without requiring direct estimation of variance.

Despite these advances, our method has several limitations. First, it relies on additional regularity assumptions to handle arbitrary client-level data heterogeneity. Second, as noted in (Shao, 2015), self-normalization yields heavier-tailed limit distributions than the Gaussian, which can produce conservative confidence intervals or reduced power in hypothesis testing. In addition, theoretically understanding how asynchronous or partial client participation affects estimation and inference is also an important direction for future work. Finally, our framework depends on a central server for aggregation and synchronization, which may not be available in fully decentralized environments. Addressing these challenges and extending the algorithm to decentralized settings remain important directions for future research.

## ACKNOWLEDGMENTS

The authors sincerely thank the anonymous reviewers, AC, and PCs for their valuable suggestions that have greatly improved the quality of our work. This work is supported by the Shanghai Engineering Research Center of Finance Intelligence (Grant No,19DZ2254600). Leheng Cai would thank to the funding supported by China Association for Science and Technology and National Natural Science Foundation of China (No. 12171269). Shuyuan Wu's research is partially supported by National Natural Science Foundation of China (No. 12401392) and China Postdoctoral Science Foundation (No. 2024M751929, No. 2024T170540).

## REPRODUCIBILITY STATEMENT

All numerical experiments and real-data analyses are fully reproducible via the code included in the supplementary materials.

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

## A  NOTATIONS AND COMPLETE ASSUMPTIONS

### A.1  NOTATIONS

### A.2  COMPLETE ASSUMPTIONS

We introduce the following necessary assumptions.

**Assumption S.1.** *For some constant $C_f > 0$, $f_k(\cdot)$, $1 \leq k \leq K$, the density function for client $k$, is uniformly bounded by $C_f$, and $\min_{1 \leq k \leq K} f_k(Q^\star) > 0$.*

**Assumption S.2.** *Define the effective step $\gamma_m = \eta_m E_m$, which is non-increasing in $m$ and satisfies that $\sum_{m=1}^{\infty} \gamma_m^2 < \infty$, $\sum_{m=1}^{\infty} \gamma_m = \infty$, and $(\gamma_m - \gamma_{m+1})/\gamma_m = \mathcal{O}(\gamma_m)$.*

**Assumption S.3.** *The sequence $\{E_m\}_{m \geq 1}$ satisfies that*

(a) *$\{E_m\}_{m \geq 1}$ is either uniformly bounded or non-decreasing.*

(b) *There exist some $\delta > 0$ and $\nu \geq 1$ such that*

$$\limsup_{T \to \infty} \frac{1}{T^2} \left( \sum_{m=0}^{T-1} E_m^{1+\delta} \right) \left( \sum_{m=0}^{T-1} E_m^{-1-\delta} \right) < \infty, \quad \lim_{T \to \infty} \frac{1}{T^2} \left( \sum_{m=0}^{T-1} E_m \right) \left( \sum_{m=0}^{T-1} E_m^{-1} \right) = \nu.$$

(c) *Denote $t_T = \sum_{m=0}^{T-1} E_m$, satisfying*

$$\lim_{T \to \infty} \frac{\sqrt{t_T}}{T} \sum_{m=0}^{T} \gamma_m = 0, \quad \lim_{T \to \infty} \frac{\sqrt{t_T}}{T} \frac{1}{\sqrt{\gamma_T}} = 0$$

| Notations | Meanings |
|-----------|----------|
| $\mathcal{L}$ | Global Loss function |
| $\tau$ | Global quantile |
| $\mathcal{L}_{\tau_k}$ | Local loss function for client $k$ with local quantile $\tau_k$ |
| $p_k$ | the weight assigned to client $k$ |
| $\mathcal{P}_k$ | Distribution for client $k$ |
| $F_k$ | CDF for client $k$ |
| $f_k$ | Density function for client $k$ |
| $\mathcal{I}$ | Communication iteration sets |
| $t_m$ | Communication iteration time |
| $E_m$ | Interval length |
| $Q$ | Variable in loss function |
| $Q^\star$ | True or target value |
| $Q_k$ | Value at $F_k(Q^*)$ |
| $q_{t+1}^k$ | Iterations at step $t$ for client $k$ |
| $r_k$ | Response rate for for client $k$ |
| $\epsilon_k$ | Privacy budget for client $k$ |
| $\widehat{Q}_T$ | LDP federated quantile estimator |
| $\widehat{\mathcal{V}}_T$ | LDP federated quantile self-normalizer |
| $B(s)$ | Brownian motion |
| $v_{\alpha/2,g}$ | $(1-\alpha/2)$ quantile of self-normalization distribution |

Table S.1: Notations table.

Assumption S.1 is a mild and regular condition concerning the uniform boundedness of density functions; see also Cai et al. (2025a). Assumptions S.2 and S.3 require that the effective step sizes decay slowly and the communication intervals increase slowly; see also Li et al. (2022).

## B   TECHNIQUE PROOFS

**Proof of Theorem 2.1:** Note that the following recursive equation

$$q_t^k = q_{t-1}^k + \frac{1-r_k+2\tau r_k}{2r_k}\eta_{m-1}\mathbb{I}(s_t^k=1) - \frac{1+r_k-2\tau r_k}{2r_k}\eta_{m-1}\mathbb{I}(s_t^k=0),$$

is asymptotically equivalent to

$$q_t^k = q_{t-1}^k + \eta_{m-1}\frac{1}{r_k}\left\{\frac{1-r_k+2\tau r_k}{2} - \mathbb{I}(\widetilde{x}_t^k \le q_{t-1}^k)\right\},$$

where

$$\mathbb{P}(\widetilde{x}_t^k = x_t^k) = r_k, \quad \mathbb{P}(\widetilde{x}_t^k = \infty) = \mathbb{P}(\widetilde{x}_t^k = -\infty) = (1-r_k)/2.$$

Observe that the above SGD update is designed to solve the following non-DP loss function

$$\arg\min_{Q_k} \mathbb{E}_{x_k\sim\widetilde{\mathcal{P}}_k}\left\{r_k^{-1}\ell_{\widetilde{\tau}_k}(x_k, Q_k)\right\}.$$

Inspired by this, we denote

$$Q^\spadesuit := \arg\min_Q \sum_{k=1}^K p_k \mathbb{E}_{x_k\sim\widetilde{\mathcal{P}}_k}\left\{r_k^{-1}\ell_{\widetilde{\tau}_k}(x_k, Q)\right\}$$

$$= \arg\min_Q \sum_{k=1}^K \frac{p_k}{r_k}\mathbb{E}_{x_k\sim\widetilde{\mathcal{P}}_k}\left[(x_k - Q)\{\widetilde{\tau}_k - \mathbb{I}(x_k \le Q)\}\right].$$

In the following, we desire to verify that $Q^\spadesuit = Q^\star$. By the definition of the minimizer of the objective function,

$$\sum_{k=1}^K \frac{p_k}{r_k}\left\{\mathbb{P}(\widetilde{x}_t^k \le Q^\spadesuit) - \widetilde{\tau}_k\right\} = 0.$$

Since $\widetilde{\tau}_k = r_k \tau + (1 - r_k)/2$, and

$$\mathbb{P}(\widetilde{x}_t^k \leq Q^{\spadesuit}) = r_k F_k(Q^{\spadesuit}) + (1 - r_k)/2,$$

one has

$$\sum_{k=1}^{K} \left\{ p_k F_k(Q^{\spadesuit}) + p_k(1 - r_k)/(2r_k) \right\} = \sum_{k=1}^{K} \left\{ p_k(1 - r_k)/(2r_k) + p_k \tau \right\}.$$

Subtracting $\sum_{k=1}^{K} \left\{ p_k(1 - r_k)/(2r_k) \right\}$ from both sides of the equation, we obtain

$$\sum_{k=1}^{K} p_k F_k(Q^{\spadesuit}) = \sum_{k=1}^{K} p_k \tau = \tau.$$

Recall that the definition of $\tau = \sum_{k=1}^{K} p_k F_k(Q^{\star})$, we show that

$$\sum_{k=1}^{K} p_k F_k(Q^{\spadesuit}) = \sum_{k=1}^{K} p_k F_k(Q^{\star}),$$

which implies $Q^{\spadesuit} = Q^{\star}$. The proof is now completed.

**Proof of Theorem 3.1:**

Theorem 3.1 is a direct consequence of Theorem 3.2.

**Proof of Theorem 3.2:**

We follows the perturbed iterate framework that is derived by Mania et al. (2017) and also used in Li et al. (2022). Define the sequence $\bar{q}_t$ in the following way:

$$\bar{q}_t = \sum_{k=1}^{K} p_k q_t^k.$$

Define $\zeta^k = (x^k, U^k, V^k)^{\top}$, with

$$\mathbb{P}(U^k = 1) = r_k, \quad \mathbb{P}(U^k = 0) = 1 - r_k, \qquad \mathbb{P}(V^k = 1) = \mathbb{P}(V^k = 0) = 1/2.$$

For $k = 1, \ldots, K$, let $U_t^k$ and $V_t^k$ be i.i.d. copies of $U^k$ and $V^k$, respectively. Denote $\zeta_t^k = (x_t^k, U_t^k, V_t^k)^{\top}$. Define

$$\begin{aligned}
G_k(q_{t-1}^k, \zeta_t^k) &= \frac{1 + r_k - 2r_k \tau}{2r_k} \left[ \mathbf{1}\{x_t^k \leq q_{t-1}^k\} U_t^k + (1 - U_t^k)(1 - V_t^k) \right] \\
&\quad - \frac{1 - r_k + 2r_k \tau}{2r_k} \left[ \mathbf{1}\{x_t^k > q_{t-1}^k\} U_t^k + (1 - U_t^k)V_t^k \right].
\end{aligned} \tag{B.1}$$

Elementary calculations show that

$$g_k(q) := \mathbb{E}G_k(q, \zeta_t^k) = F_k(q) - \tau.$$

Define

$$g(q) = \sum_{k=1}^{K} p_k g_k(q).$$

Denote

$$\varepsilon_k(q) = G_k(q, \zeta_t^k) - g_k(q).$$

Besides,

$$\mathbb{E}\left(\varepsilon_k^2(q_{t-1}^k) | \mathcal{F}_{t-1}\right) = \frac{1 - r_k^2 \{2F_k(q_{t-1}^k) - 1\}^2}{4r_k^2}.$$

By definition, for $t_m \le t < t_{m+1} - 1$, we have

$$q_{t+1}^k = \bar{q}_{t_m} - \eta_m \sum_{i=t_m}^{t} G_k(q_i^k, \zeta_i^k).$$

Define $s_m = \bar{q}_{t_m} - Q^\star$, and recall that $E_m = t_{m+1} - t_m$ and $\gamma_m = \eta_m E_m$. Elementary Iteration from $t = t_m$ to $t_{m+1} - 1$ yields

$$s_{m+1} = s_m - \eta_m \sum_{t=t_m}^{t_{m+1}-1} \sum_{k=1}^{K} p_k G_k(q_{t-1}^k, \zeta_t^k) = s_m - \gamma_m \nu_m,$$

in which

$$\nu_m = \frac{1}{E_m} \sum_{t=t_m}^{t_{m+1}-1} \sum_{k=1}^{K} p_k G_k(q_{t-1}^k, \zeta_t^k).$$

We define

$$h_m := \frac{1}{E_m} \sum_{t=t_m}^{t_{m+1}-1} \sum_{k=1}^{K} p_k G_k\left(\bar{q}_{t_m}; \zeta_t^k\right),$$

and further decompose that

$$\nu_m = Gs_m + (g(\bar{q}_{t_m}) - Gs_m) + (h_m - g(\bar{q}_{t_m})) + (\nu_m - h_m)$$
$$:= Gs_m + r_m + \varepsilon_m + \delta_m,$$

where $G = \sum_{k=1}^{K} p_k f_k(Q^\star)$ is the Hessian at $Q^\star$. It then follows that

$$s_{m+1} = (1 - \gamma_m G) s_m - \gamma_m (r_m + \varepsilon_m + \delta_m) := B_m s_m - \gamma_m U_m, \tag{B.2}$$

where $B_m := 1 - \gamma_m G$ and $U_m := r_m + \varepsilon_m + \delta_m$ for short. Recurring (B.2) gives

$$s_{m+1} = \left(\prod_{j=0}^{m} B_j\right) s_0 - \sum_{j=0}^{m} \left(\prod_{i=j+1}^{m} B_i\right) \gamma_j U_j.$$

Here, we use the convention that $\prod_{i=m+1}^{m} B_i = 1$ for any $m \ge 0$. Recall the definition that

$$h(r, T) = \max \left\{ n \in \mathbb{Z}_+ \ \middle| \ r \sum_{m=1}^{T} \frac{1}{E_m} \ge \sum_{m=1}^{n} \frac{1}{E_m} \right\}.$$

Hence,

$$\frac{\sqrt{t_T}}{T} \sum_{m=0}^{h(r,T)} s_{m+1} = \frac{\sqrt{t_T}}{T} \sum_{m=0}^{h(r,T)} \left[ \left(\prod_{j=0}^{m} B_j\right) s_0 - \sum_{j=0}^{m} \left(\prod_{i=j+1}^{m} B_i\right) \gamma_j U_j \right]$$

$$= \frac{\sqrt{t_T}}{T} \sum_{m=0}^{h(r,T)} \left(\prod_{j=0}^{m} B_j\right) s_0 - \frac{\sqrt{t_T}}{T} \sum_{j=0}^{h(r,T)} \sum_{m=j}^{h(r,T)} \left(\prod_{i=j+1}^{m} B_i\right) \gamma_j U_j.$$

For any $n \ge j$, define $A_j^n$ as

$$A_j^n = \sum_{l=j}^{n} \left(\prod_{i=j+1}^{l} B_i\right) \gamma_j.$$

With the notation of $A_j^n$, we can rewrite that

$$\frac{\sqrt{t_T}}{T} \sum_{m=0}^{h(r,T)} s_{m+1} = \frac{\sqrt{t_T}}{T\gamma_0} A_0^{h(r,T)} B_0 s_0 - \frac{\sqrt{t_T}}{T} \sum_{m=0}^{h(r,T)} A_m^{h(r,T)} U_m.$$

Since $U_m = r_m + \varepsilon_m + \delta_m$, then

$$\frac{\sqrt{t_T}}{T} \sum_{m=0}^{h(r,T)} s_{m+1} + \frac{\sqrt{t_T}}{T} \sum_{m=0}^{h(r,T)} G^{-1} \varepsilon_m = \frac{\sqrt{t_T}}{T\gamma_0} A_0^{h(r,T)} B_0 s_0 - \frac{\sqrt{t_T}}{T} \sum_{m=0}^{h(r,T)} A_m^{h(r,T)} \left( r_m + \delta_m \right)$$

$$- \frac{\sqrt{t_T}}{T} \sum_{m=0}^{h(r,T)} \left( A_m^T - G^{-1} \right) \varepsilon_m$$

$$- \frac{\sqrt{t_T}}{T} \sum_{m=0}^{h(r,T)} \left( A_m^{h(r,T)} - A_m^T \right) \varepsilon_m$$

$$=: \mathcal{T}_0 - \mathcal{T}_1 - \mathcal{T}_2 - \mathcal{T}_3.$$

To complete the proof, we first investigate the partial-sum asymptotic behavior of

$$\frac{\sqrt{t_T}}{T} \sum_{m=0}^{h(r,T)} G^{-1} \varepsilon_m,$$

and then show that the four separate terms: $\sup_{r \in [0,1]} \|\mathcal{T}_0\|, \sup_{r \in [0,1]} \|\mathcal{T}_1\|, \sup_{r \in [0,1]} \|\mathcal{T}_2\|$, and $\sup_{r \in [0,1]} \|\mathcal{T}_4\|$ are $o_{\mathbb{P}}(1)$, respectively.

We aim to follow the proof of Theorem 4.2 in Li et al. (2022). However, we find that the average smoothness condition in their Assumption 3.1 is not satisfied. Due to the presence of the indicator function in (B.1), we only have

$$\sqrt{\mathbb{E}\left\{ G_k(x, \zeta_t^k) - G_k(y, \zeta_t^k) \right\}^2} \lesssim |x - y|^{1/2}. \tag{B.3}$$

Upon close examination of their proof, we find that this condition is crucial in the proof of their key Lemma B.2.

In the following, we re-establish the proof of

$$\mathbb{E}|\bar{q}_{t_m} - Q^\star|^2 \lesssim \gamma_m, \quad \bar{q}_{t_m} \xrightarrow{a.s.} Q^*.$$

under the condition given in (B.3). Consider that

$$\mathbb{E}\left( \left| q_{t+1}^k - \bar{q}_{t_m} \right| \mid \mathcal{F}_{t_m} \right) = \eta_m \mathbb{E}\left( \left| \sum_{i=t_m}^{t} G_k(q_i^k, \zeta_i^k) \right| \mid \mathcal{F}_{t_m} \right)$$

$$\leq \eta_m \sum_{i=t_m}^{t} \mathbb{E}\left( |G_k(q_i^k, \zeta_i^k)| \mid \mathcal{F}_{t_m} \right)$$

$$\lesssim \eta_m \sum_{i=t_m}^{t} \left( 1 + |q_i^k - \bar{q}_{t_m}| + |\bar{q}_{t_m} - Q^\star| \right),$$

where the last inequality holds by the following fact

$$\mathbb{E}\left( G_k^2(q_i^k, \zeta_i^k) \mid \mathcal{F}_i \right) = \mathbb{E}\left( \left| G_k(q_i^k, \zeta_i^k) - g_k(q_i^k) \right|^2 \mid \mathcal{F}_i \right) + g_k^2(q_i^k)$$

$$\leq \mathbb{E}\left( \left| G_k(q_i^k, \zeta_i^k) - g_k(q_i^k) \right|^2 \mid \mathcal{F}_i \right) + 2|g_k(q_i^k) - g_k(Q^\star)|^2 + 2g_k^2(Q^\star)$$

$$\lesssim \left\{ 1 + 2g_k^2(Q^\star) \right\} + |q_i^k - Q^\star|^2$$

$$\lesssim 1 + |q_i^k - \bar{q}_{t_m}|^2 + |\bar{q}_{t_m} - Q^\star|^2.$$

Define

$$V_t = \sum_{k=1}^{K} p_k \mathbb{E}\left( |q_t^k - \bar{q}_{t_m}| \mid \mathcal{F}_{t_m} \right).$$

Hence,

$$V_{t+1} \lesssim \eta_m \sum_{i=t_m}^{t} (1 + |\bar{q}_{t_m} - Q^\star| + V_i),$$

which further implies that (since $V_{t_m} = 0$)

$$\frac{1}{E_m} \sum_{t=t_m}^{t_{m+1}-1} V_t = \frac{1}{E_m} \sum_{t=t_m}^{t_{m+1}-2} V_{t+1} \lesssim \frac{\eta_m}{E_m} \sum_{t=t_m}^{t_{m+1}-2} \sum_{i=t_m}^{t} (1 + |\bar{q}_{t_m} - Q^\star| + V_i)$$

$$= \frac{\eta_m}{E_m} \sum_{t=t_m}^{t_{m+1}-2} (t_{m+1} - t - 1)(1 + |\bar{q}_{t_m} - Q^\star| + V_t)$$

$$\lesssim \eta_m \sum_{t=t_m}^{t_{m+1}-2} (1 + |\bar{q}_{t_m} - Q^\star| + V_t)$$

$$\lesssim \eta_m E_m \left(1 + |\bar{q}_{t_m} - Q^\star| + \frac{1}{E_m} \sum_{t=t_m}^{t_{m+1}-1} V_t\right).$$

Denote by $\gamma_m = \eta_m E_m$. It follows that

$$\frac{1}{E_m} \sum_{t=t_m}^{t_{m+1}-1} V_t \lesssim \gamma_m (1 + |\bar{q}_{t_m} - Q^\star|). \tag{B.4}$$

Let $\mathcal{G}_k(\cdot)$ denote an antiderivative of $g_k(\cdot)$, and $\mathcal{G}(\cdot) = \sum_{k=1}^{K} p_k \mathcal{G}_k(\cdot)$. Let $\Delta_m = \mathcal{G}(\bar{q}_{t_m}) - \mathcal{G}(Q^\star)$. The equation (17) in Li et al. (2022) shows that for some constant $L > 0$,

$$\mathbb{E}\{\mathcal{G}(\bar{q}_{t_m+1})|\mathcal{F}_{t_m}\} \leq \mathcal{G}(\bar{q}_{t_m}) - \gamma_m/2 |\nabla \mathcal{G}(\bar{q}_{t_m})|^2 + \gamma_m^2 L \mathbb{E}(h_m^2|\mathcal{F}_{t_m})$$
$$+ (\gamma_m/2 + \gamma_m^2 L) \mathbb{E}(\delta_m^2|\mathcal{F}_{t_m}),$$

where

$$h_m = \frac{1}{E_m} \sum_{t=t_m}^{t_{m+1}-1} \sum_{k=1}^{K} p_k \nabla G_k(\bar{q}_{t_m}^k, \zeta_t^k), \quad \delta_m = \frac{1}{E_m} \sum_{t=t_m}^{t_{m+1}-1} \sum_{k=1}^{K} p_k \nabla G_k(\bar{q}_t^k, \zeta_t^k).$$

Lemma B.9 of Li et al. (2022) obtains that

$$\mathbb{E}(h_m^2|\mathcal{F}_{t_m}) \leq |\nabla \mathcal{G}(\bar{q}_{t_m})|^2 + \frac{C_1}{E_m} + \frac{C_2}{E_m} |\bar{q}_{t_m} - Q^\star|^2.$$

Notice that

$$\mathbb{E}\left\{(G_k(q_{t-1}^k, \zeta_t^k) - G_k(\bar{q}_{t_m}, \zeta_t^k))^2 |\mathcal{F}_{t_m}\right\} \lesssim \mathbb{E}\left(|q_t^k - \bar{q}_{t_m}| \,|\mathcal{F}_{t_m}\right).$$

Thus,

$$\mathbb{E}(\delta_m^2|\mathcal{F}_{t_m}) \lesssim \frac{1}{E_m} \sum_{t=t_m}^{t_{m+1}-1} \sum_{k=1}^{K} p_k \mathbb{E}\left\{(G_k(q_{t-1}^k, \zeta_t^k) - G_k(\bar{q}_{t_m}, \zeta_t^k))^2 |\mathcal{F}_{t_m}\right\}$$

$$\lesssim \frac{1}{E_m} \sum_{t=t_m}^{t_{m+1}-1} \sum_{k=1}^{K} p_k \mathbb{E}\left(|q_t^k - \bar{q}_{t_m}| \,|\mathcal{F}_{t_m}\right)$$

$$\lesssim \gamma_m (1 + |\bar{q}_{t_m} - Q^\star|),$$

where the last inequality holds by (B.4). Therefore, we obtain that

$$\mathbb{E}(\Delta_{m+1}|\mathcal{F}_{t_m}) \leq \Delta_m - \gamma_m/2 |\nabla \mathcal{G}(\bar{q}_{t_m})|^2 + \gamma_m^2 L \left\{|\nabla \mathcal{G}(\bar{q}_{t_m})|^2 + \frac{C_1}{E_m} + \frac{C_2}{E_m} |\bar{q}_{t_m} - Q^\star|^2\right\}$$
$$+ (\gamma_m/2 + \gamma_m^2 L) \gamma_m (C_1 + C_2 |\bar{q}_{t_m} - Q^*|)$$

$$\leq (1 - c_1\gamma_m + c_2\gamma_m^2)\Delta_m + \left(c_3 + c_4\Delta_m^{1/2}\right)\gamma_m^2.$$

Since we assume that the parameter space is uniformly bounded, it entails that $\Delta_m$ is also uniformly bounded. Thus, we have

$$\mathbb{E}(\Delta_{m+1}|\mathcal{F}_{t_m}) \leq (1 - c_1\gamma_m + c_2\gamma_m^2)\Delta_m + (c_3 + c_5)\gamma_m^2.$$

Apply Robbins-Siegmund theorem in Robbins & Siegmund (1971) to obtain $\bar{q}_{t_m} \to Q^\star$ almost surely. Lemma A.10 in Su & Zhu (2018) states that for any positive constants $c_1, c_2$, if $\gamma_m = o(1)$, $\gamma_{m-1}/\gamma_m = 1 + o(\gamma_m)$, and $B_m$ is a positive sequence, satisfying

$$B_m \leq \frac{\gamma_{m-1}(1 - c_1\gamma_m)}{\gamma_m}B_{m-1} + c_2\gamma_m,$$

then $\sup_m B_m < \infty$. Using this lemma, we immediately obtain that for some positive constant $C > 0$,

$$\sup_{m \geq 1} \frac{\mathbb{E}\Delta_m}{\gamma_{m-1}} < C,$$

which entails that

$$\mathbb{E}|\bar{q}_{t_m} - Q^\star|^2 \lesssim \mathbb{E}\Delta_m \lesssim \gamma_{m-1} \lesssim \gamma_m\{1 + o(\gamma_m)\} \lesssim \gamma_m.$$

To demonstrate that our setting satisfies Assumption 3.2 of Li et al. (2022), we define

$$S_k := \mathbb{E}\varepsilon_k^2(Q^\star) = \frac{1 - r_k^2\{2F_k(Q^\star) - 1\}^2}{4r_k^2} = \frac{1 - r_k^2\{2Q_k - 1\}^2}{4r_k^2}.$$

Hence,

$$\left|\mathbb{E}\left(\varepsilon_k^2(q_{t-1}^k)|\mathcal{F}_{t-1}\right) - \mathbb{E}\varepsilon_k^2(Q^\star)\right| \lesssim |q_{t-1}^k - Q^\star|,$$

satisfying Assumption 3.2 in Li et al. (2022). Assumptions 3.3 and 3.4 of Li et al. (2022) are the same as our Assumptions 3-4.

By closely examining the proof of Li et al. (2022), we find that their Assumption 3.1 is used exclusively to establish the $\mathcal{L}^2$ convergence rate and the almost sure convergence result in their Lemma B.2. The appearance of Assumption 1 in the conditions of their Lemma B.3, which derivs the functional weak convergence, is solely for enabling the application of Lemma B.2.

In our analysis, however, the key bound $\mathbb{E}|\bar{q}_{t_m} - Q^\star|^2 \lesssim \gamma_m$ is proved directly under equation (B.3). This bound fully replaces the role played by Lemma B.2 in Li et al. (2022). Consequently, Assumption 3.1 of Li et al. (2022) is not needed in our theoretical development beyond this step, and the functional central limit theorem follows without invoking that assumption.

Recall that

$$\varepsilon_m = \frac{1}{E_m} \sum_{t=t_m}^{t_{m+1}-1} \sum_{k=1}^K p_k \left\{G_k(\bar{q}_{t_m}, \zeta_t^k) - g(\bar{q}_{t_m})\right\}.$$

Define $U_T^2 = \sum_{m=1}^T \mathbb{E}(\varepsilon_m^2|\mathcal{F}_{t_m})$ and

$$\Xi_T(r) := U_T^{-1}\left\{\sum_{m=1}^i \varepsilon_m + \frac{(rU_T^2 - U_i^2)\varepsilon_{i+1}}{U_{i+1}^2 - U_i^2}\right\} \quad U_i^2 \leq rU_T^2 < U_{i+1}^2.$$

Then,

$$\frac{\sqrt{t_T}}{T}\sum_{m=1}^{h(r,T)} \varepsilon_m = \frac{\sqrt{t_T}}{T}U_T\Xi_T\left(\frac{U_{h(r,T)}^2}{U_T^2}\right).$$

Following Lemma B.16 in Li et al. (2022), one obtains that

$$\sup_{r \in [0,1]} \left|\frac{\sqrt{t_T}}{T}U_T\Xi_T\left(\frac{U_{h(r,T)}^2}{U_T^2}\right) - \frac{\sqrt{t_T}}{T}U_T\Xi_T(r)\right| \xrightarrow{\mathbb{P}} 0. \tag{B.5}$$

By evaluating the conditions in Lemma B.13 of Li et al. (2022), the invariance principles in the martingale CLT yileds that $\Xi_T(r) \xrightarrow{d} B(r)$ in $\mathcal{C}[0,1]$, which is followed by

$$\frac{\sqrt{t_T}}{T} U_T \Xi_T(r) \xrightarrow{d} \frac{\sqrt{\nu \sum_{k=1}^K p_k^2 \left\{ r_k^{-2} - (2Q_k - 1)^2 \right\}}}{2 \sum_{k=1}^K p_k f_k(Q^\star)} B(r).$$

Hus, Theorem A.2 of Hall & Heyde (1980) implies that $\left\{ \sqrt{t_T} U_T \Xi_T(\cdot)/T \right\}_{T \geq 1}$ is tight in $\mathcal{C}[0,1]$. Using the tightness of $\left\{ \sqrt{t_T} U_T \Xi_T(\cdot)/T \right\}_{T \geq 1}$ and (B.5), we could obtain the tightness of

$$\left\{ \frac{\sqrt{t_T}}{T} U_T \Xi_T \left( \frac{U_{h(r,T)}^2}{U_T^2} \right) \right\}_{T \geq 1}$$

following the arguments in the proof of Lemma B.16 of Li et al. (2022). Therefore, we could follow the arguments in the proof of Theorem 4.2 in Li et al. (2022) to complete the proof the functional CLT.

## C  ADDITIONAL EXPERIMENTS AND DISCUSSIONS

In this section, we report additional simulation results that complement the main experiments in Section 4. Unless otherwise specified, all settings are identical to those in Section 4.1.

### C.1  OTHER RESULTS IN SECTION 4

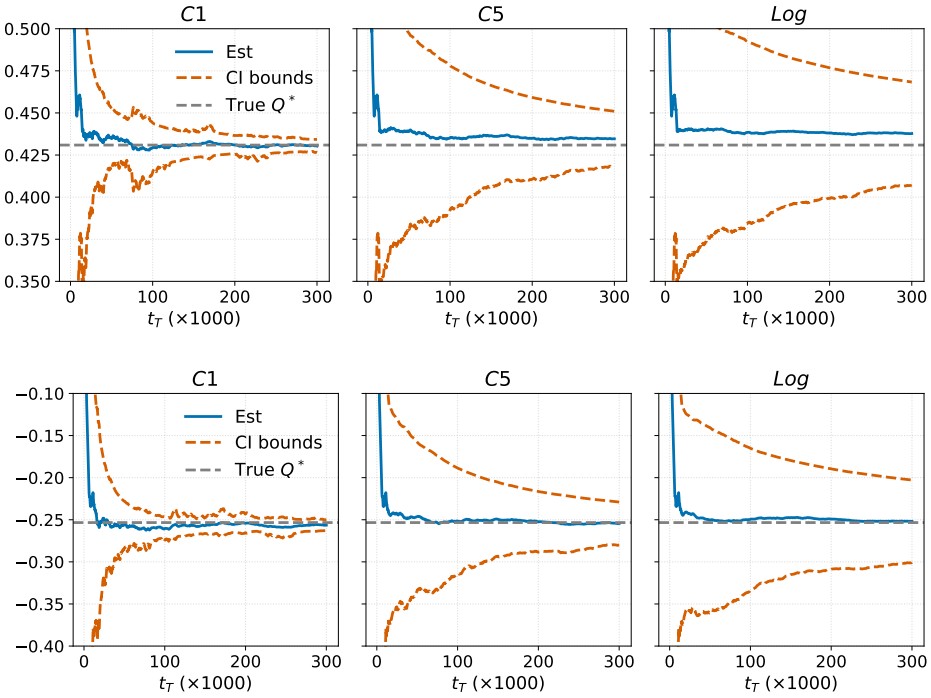

Figure S.1: Sample trajectories of the iterative estimator and corresponding confidence intervals under heterogeneous distributions (Hete L, with $r_k = 0.9$ and $\tau = 0.5$, upper panel) and heterogeneous quantile levels ($\tau_{\text{low}}$, with heterogeneous response rates, lower panel). The horizontal dotted line indicates the true quantile value $Q^*$.

**Efficiency–Accuracy Trade-off.** We first quantitatively study the computation and communication time of the proposed method. Specifically, for each simulation run, we record the wall-clock

| Quantile ($\tau$) | Rate ($r$) | C1 | C5 | Log | DP-SGD (C1) | DC | Single |
|---|---|---|---|---|---|---|---|
| | | | | $t_T = 10000$ | | | |
| 0.5 | 0.25 | 0.950(0.0160) | 0.979(0.0250) | 0.978(0.0324) | 0.952(0.0242) | 0.802(0.3197) | 0.926(0.0179) |
| 0.5 | hetero | 0.953(0.0086) | 0.994(0.0124) | 0.993(0.0168) | 0.956(0.0125) | 0.999(0.0678) | 0.942(0.0075) |
| 0.5 | 0.9 | 1.000(0.0027) | 1.000(0.0057) | 1.000(0.0097) | 0.987(0.0044) | 1.000(0.0216) | 0.954(0.0047) |
| $\tau_{\text{low}}$ | 0.25 | 0.929(0.0196) | 0.976(0.0270) | 0.978(0.0349) | 0.911(0.0309) | 0.740(0.3403) | 0.930(0.0194) |
| $\tau_{\text{low}}$ | hetero | 0.923(0.0118) | 0.983(0.0131) | 0.983(0.0174) | 0.896(0.0181) | 0.999(0.0735) | 0.960(0.0083) |
| $\tau_{\text{low}}$ | 0.9 | 0.969(0.0055) | 0.999(0.0047) | 1.000(0.0081) | 0.922(0.0074) | 1.000(0.0231) | 0.954(0.0054) |
| $\tau_{\text{high}}$ | 0.25 | 0.956(0.0205) | 0.972(0.0310) | 0.971(0.0399) | 0.944(0.0313) | 0.729(0.3768) | 0.944(0.0213) |
| $\tau_{\text{high}}$ | hetero | 0.956(0.0109) | 0.986(0.0145) | 0.988(0.0198) | 0.959(0.0160) | 0.997(0.0839) | 0.954(0.0089) |
| $\tau_{\text{high}}$ | 0.9 | 0.999(0.0049) | 0.997(0.0046) | 1.000(0.0068) | 0.989(0.0059) | 1.000(0.0259) | 0.974(0.0059) |
| | | | | $t_T = 50000$ | | | |
| 0.5 | 0.25 | 0.960(0.0070) | 0.987(0.0101) | 0.991(0.0144) | 0.955(0.0104) | 0.977(0.0787) | 0.958(0.0071) |
| 0.5 | hetero | 0.975(0.0037) | 0.986(0.0052) | 0.994(0.0077) | 0.975(0.0055) | 1.000(0.0156) | 0.970(0.0030) |
| 0.5 | 0.9 | 1.000(0.0010) | 1.000(0.0016) | 1.000(0.0029) | 0.995(0.0019) | 1.000(0.0051) | 0.950(0.0020) |
| $\tau_{\text{low}}$ | 0.25 | 0.913(0.0104) | 0.973(0.0119) | 0.988(0.0154) | 0.878(0.0171) | 0.973(0.0863) | 0.946(0.0079) |
| $\tau_{\text{low}}$ | hetero | 0.819(0.0076) | 0.942(0.0070) | 0.977(0.0084) | 0.786(0.0116) | 0.999(0.0166) | 0.952(0.0034) |
| $\tau_{\text{low}}$ | 0.9 | 0.987(0.0042) | 0.988(0.0018) | 0.999(0.0022) | 0.854(0.0054) | 1.000(0.0054) | 0.962(0.0021) |
| $\tau_{\text{high}}$ | 0.25 | 0.951(0.0107) | 0.986(0.0163) | 0.980(0.0250) | 0.910(0.0183) | 0.976(0.0983) | 0.950(0.0089) |
| $\tau_{\text{high}}$ | hetero | 0.967(0.0057) | 0.995(0.0103) | 0.988(0.0164) | 0.924(0.0110) | 1.000(0.0189) | 0.946(0.0039) |
| $\tau_{\text{high}}$ | 0.9 | 1.000(0.0012) | 1.000(0.0043) | 1.000(0.0083) | 0.995(0.0027) | 1.000(0.0059) | 0.946(0.0024) |

Table S.2: Empirical coverage probabilities at the 95% nominal level (mean absolute errors $\downarrow$) under varying quantile levels and response rates, with different $t_T$ and fixed $K = 10$ clients and data generated from $\mathcal{C}(0, 1)$. In Case $\tau_{\text{low}}$, each client uses a unique quantile level $\tau_k$ ranging uniformly from $[0.3, 0.5]$; in Case $\tau_{\text{high}}$, $\tau_k$ is ranging from $[0.5, 0.8]$. "hetero" indicates client-specific truthful response rates $r_k$ range uniformly from $[0.25, 0.9]$.

| Quantile ($\tau$) | Rate ($r$) | C1 | C5 | Log | DP-SGD (C1) | DC | Single |
|---|---|---|---|---|---|---|---|
| | | | | $t_T = 10000$ | | | |
| 0.3 | 0.25 | 0.947(0.0271) | 0.981(0.0400) | 0.967(0.0545) | 0.925(0.0423) | 0.546(0.6111) | 0.946(0.0339) |
| 0.3 | hetero | 0.957(0.0138) | 0.994(0.0231) | 0.992(0.0311) | 0.942(0.0223) | 0.682(0.3476) | 0.948(0.0137) |
| 0.3 | 0.9 | 1.000(0.0039) | 1.000(0.0170) | 1.000(0.0254) | 0.994(0.0066) | 0.070(0.3084) | 0.972(0.0083) |
| 0.5 | 0.25 | 0.942(0.0225) | 0.983(0.0298) | 0.977(0.0389) | 0.935(0.0314) | 0.968(0.3335) | 0.936(0.0264) |
| 0.5 | hetero | 0.946(0.0132) | 0.970(0.0155) | 0.981(0.0196) | 0.951(0.0170) | 1.000(0.0757) | 0.942(0.0109) |
| 0.5 | 0.9 | 0.976(0.0095) | 0.994(0.0053) | 0.998(0.0064) | 0.953(0.0100) | 0.968(0.0519) | 0.968(0.0071) |
| 0.8 | 0.25 | 0.935(0.0450) | 0.958(0.0729) | 0.960(0.0963) | 0.938(0.0677) | 0.087(1.0105) | 0.950(0.0552) |
| 0.8 | hetero | 0.956(0.0232) | 0.963(0.0370) | 0.971(0.0457) | 0.963(0.0343) | 0.557(0.5459) | 0.968(0.0210) |
| 0.8 | 0.9 | 0.985(0.0171) | 0.836(0.0269) | 0.933(0.0267) | 0.971(0.0165) | 0.161(0.4245) | 0.982(0.0122) |
| | | | | $t_T = 50000$ | | | |
| 0.3 | 0.25 | 0.902(0.0151) | 0.979(0.0165) | 0.988(0.0225) | 0.863(0.0253) | 0.798(0.3556) | 0.958(0.0137) |
| 0.3 | hetero | 0.835(0.0102) | 0.975(0.0081) | 0.994(0.0108) | 0.831(0.0164) | 0.023(0.3026) | 0.948(0.0059) |
| 0.3 | 0.9 | 0.984(0.0056) | 1.000(0.0034) | 1.000(0.0079) | 0.869(0.0072) | 0.000(0.2951) | 0.938(0.0037) |
| 0.5 | 0.25 | 0.935(0.0100) | 0.983(0.0127) | 0.986(0.0180) | 0.926(0.0144) | 1.000(0.0910) | 0.958(0.0108) |
| 0.5 | hetero | 0.942(0.0063) | 0.981(0.0064) | 0.995(0.0093) | 0.947(0.0081) | 0.853(0.0496) | 0.954(0.0045) |
| 0.5 | 0.9 | 0.998(0.0049) | 0.999(0.0017) | 1.000(0.0039) | 0.959(0.0052) | 0.477(0.0481) | 0.948(0.0029) |
| 0.8 | 0.25 | 0.926(0.0247) | 0.991(0.0344) | 0.986(0.0514) | 0.887(0.0437) | 0.522(0.5824) | 0.952(0.0202) |
| 0.8 | hetero | 0.925(0.0149) | 0.990(0.0231) | 0.990(0.0371) | 0.890(0.0278) | 0.079(0.4077) | 0.950(0.0086) |
| 0.8 | 0.9 | 1.000(0.0038) | 1.000(0.0107) | 1.000(0.0197) | 0.979(0.0077) | 0.001(0.3822) | 0.968(0.0049) |

Table S.3: Empirical coverage probabilities at the 95% nominal level (mean absolute errors $\downarrow$) under heterogeneous distributions for different $t_T$. The number of clients $K$ is fixed at 10. Data for each client $k$ are independently generated from $\mathcal{C}(\mu_k, 1)$, where $\mu_k \sim \mathcal{N}(0, 1)$. "hetero" indicates client-specific truthful response rates $r_k$ range uniformly from $[0.25, 0.9]$.

| Quantile ($\tau$) | Rate ($r$) | C1 | C5 | Log |
|---|---|---|---|---|
| | | $T = 5000$ | | |
| 0.5 | 0.25 | 0.954(0.0189) | 0.974(0.0129) | 0.986(0.0112) |
| 0.5 | hetero | 0.959(0.0103) | 0.976(0.0065) | 0.995(0.0052) |
| 0.5 | 0.9 | 0.999(0.0033) | 1.000(0.0040) | 1.000(0.0026) |
| $\tau_{\text{low}}$ | 0.25 | 0.957(0.0200) | 0.974(0.0137) | 0.991(0.0116) |
| $\tau_{\text{low}}$ | hetero | 0.957(0.0108) | 0.977(0.0067) | 0.993(0.0053) |
| $\tau_{\text{low}}$ | 0.9 | 1.000(0.0033) | 1.000(0.0040) | 1.000(0.0029) |
| $\tau_{\text{high}}$ | 0.25 | 0.956(0.0212) | 0.975(0.0128) | 0.993(0.0123) |
| $\tau_{\text{high}}$ | hetero | 0.961(0.0112) | 0.984(0.0062) | 0.996(0.0056) |
| $\tau_{\text{high}}$ | 0.9 | 0.998(0.0037) | 0.997(0.0028) | 1.000(0.0031) |
| | | $T = 10000$ | | |
| 0.5 | 0.25 | 0.949(0.0133) | 0.968(0.0078) | 0.987(0.0061) |
| 0.5 | hetero | 0.963(0.0071) | 0.978(0.0037) | 0.991(0.0030) |
| 0.5 | 0.9 | 0.995(0.0023) | 0.999(0.0020) | 0.999(0.0014) |
| $\tau_{\text{low}}$ | 0.25 | 0.947(0.0136) | 0.972(0.0078) | 0.984(0.0064) |
| $\tau_{\text{low}}$ | hetero | 0.962(0.0072) | 0.985(0.0038) | 0.983(0.0033) |
| $\tau_{\text{low}}$ | 0.9 | 0.999(0.0020) | 1.000(0.0016) | 0.967(0.0018) |
| $\tau_{\text{high}}$ | 0.25 | 0.939(0.0145) | 0.974(0.0086) | 0.985(0.0066) |
| $\tau_{\text{high}}$ | hetero | 0.968(0.0076) | 0.988(0.0043) | 0.985(0.0032) |
| $\tau_{\text{high}}$ | 0.9 | 0.996(0.0023) | 0.999(0.0031) | 0.996(0.0014) |

Table S.4: Empirical coverage probabilities at the 95% nominal level (mean absolute errors ↓) under varying quantile levels and response rates, with different $T$ and fixed $K = 10$ clients and data generated from $\mathcal{N}(0, 1)$. In Case $\tau_{\text{low}}$, each client uses a unique quantile level $\tau_k$ ranging uniformly from $[0.3, 0.5]$; in Case $\tau_{\text{high}}$, $\tau_k$ is ranging from $[0.5, 0.8]$. "hetero" indicates client-specific truthful response rates $r_k$ range uniformly from $[0.25, 0.9]$.

time, including both computation and communication components, and then average the results over 1,000 repetitions. We consider different communication strategies for the proposed estimators as well as the competing baselines. Figure S.2 reports the results for a representative setting with quantile level $\tau_k \equiv \tau = 0.5$, truthful response rate $r_k \equiv r = 0.5$, data generated from a standard normal distribution, and total sample size $t_T = 10,000$. We obtain the following interesting findings. First, for the proposed methods, as the communication frequency increases, the MAE decreases, but both computation time and communication cost naturally grow, demonstrating a clear efficiency–accuracy trade-off. In addition, comparing across different methods, the proposed method attains a communication cost comparable to DP-SGD while requiring noticeably less computation time. The DC method incurs the lowest overall cost because it performs only a single aggregation step; however, it also exhibits the largest MAE and substantial under-coverage in several heterogeneous settings, as reported in Section 4.

**Partial Participation.** We further examine partial client participation to evaluate the proposed method in more realistic federated environments. In this setting, at each communication round, only 5 out of the 10 clients are randomly selected to participate, while all other configurations remain identical to those in Section 4.1. The results are reported in Tables S.6 and S.7. We observe that the empirical coverage probabilities remain close to the nominal 95% level, and the MAE increases only mildly compared with the full-participation case. These findings indicate that the proposed estimator is robust to partial and asynchronous client participation.

## C.2 SENSITIVITY ANALYSIS

We begin by examining how the proposed method responds to different truthful response rates $r$ and step-size schedules. Throughout this subsection, we adopt the normal design with $X \sim \mathcal{N}(0, 1)$ and target quantile level $\tau_k \equiv \tau = 0.5$, and we consider $E'_m = 5$ and $t_T \in \{10,000, 50,000\}$. All other settings follow Section 4.1.

| Quantile ($\tau$) | Rate ($r$) | C1 | C5 | Log |
|---|---|---|---|---|
| | | Hete L — $T = 5000$ | | |
| 0.3 | 0.25 | 0.942(0.0271) | 0.960(0.0168) | 0.975(0.0151) |
| 0.3 | hetero | 0.962(0.0131) | 0.966(0.0086) | 0.987(0.0067) |
| 0.3 | 0.9 | 0.998(0.0043) | 0.959(0.0063) | 1.000(0.0033) |
| 0.5 | 0.25 | 0.954(0.0254) | 0.973(0.0154) | 0.990(0.0153) |
| 0.5 | hetero | 0.963(0.0120) | 0.981(0.0072) | 0.991(0.0071) |
| 0.5 | 0.9 | 0.992(0.0042) | 0.998(0.0032) | 1.000(0.0034) |
| 0.8 | 0.25 | 0.954(0.0375) | 0.982(0.0242) | 0.998(0.0248) |
| 0.8 | hetero | 0.968(0.0181) | 0.988(0.0109) | 0.998(0.0116) |
| 0.8 | 0.9 | 0.985(0.0108) | 0.999(0.0070) | 0.982(0.0094) |
| | | Hete L — $T = 10000$ | | |
| 0.3 | 0.25 | 0.958(0.0184) | 0.966(0.0102) | 0.981(0.0083) |
| 0.3 | hetero | 0.949(0.0096) | 0.965(0.0050) | 0.979(0.0040) |
| 0.3 | 0.9 | 1.000(0.0029) | 0.979(0.0022) | 0.867(0.0036) |
| 0.5 | 0.25 | 0.950(0.0165) | 0.974(0.0094) | 0.985(0.0085) |
| 0.5 | hetero | 0.952(0.0085) | 0.976(0.0045) | 0.991(0.0039) |
| 0.5 | 0.9 | 0.996(0.0025) | 0.985(0.0018) | 1.000(0.0016) |
| 0.8 | 0.25 | 0.966(0.0237) | 0.983(0.0163) | 0.990(0.0149) |
| 0.8 | hetero | 0.962(0.0122) | 0.988(0.0088) | 0.974(0.0090) |
| 0.8 | 0.9 | 0.990(0.0042) | 0.997(0.0087) | 0.645(0.0095) |
| | | Hete D — $T = 5000$ | | |
| 0.5 | 0.25 | 0.954(0.0195) | 0.974(0.0129) | 0.987(0.0109) |
| 0.5 | hetero | 0.965(0.0098) | 0.974(0.0075) | 0.993(0.0049) |
| 0.5 | 0.9 | 1.000(0.0037) | 0.989(0.0060) | 1.000(0.0026) |
| | | Hete D — $T = 10000$ | | |
| 0.5 | 0.25 | 0.949(0.0132) | 0.968(0.0078) | 0.982(0.0064) |
| 0.5 | hetero | 0.966(0.0069) | 0.973(0.0039) | 0.972(0.0034) |
| 0.5 | 0.9 | 1.000(0.0023) | 0.999(0.0014) | 0.966(0.0023) |

Table S.5: Empirical coverage probabilities at the 95% nominal level (mean absolute errors $\downarrow$) under heterogeneous distributions for different $T$. The number of clients $K$ is fixed at 10. In Hete L, data for each client $k$ are independently generated from $\mathcal{N}(\mu_k, 1)$, where $\mu_k \sim \mathcal{N}(0, 1)$. In Hete D, data are generated from $\mathcal{N}(0, 1)$, $\mathcal{U}(-1, 1)$, and $\mathcal{C}(0, 1)$ across different clients. "hetero" indicates client-specific truthful response rates $r_k$ range uniformly from $[0.25, 0.9]$.

First, we conduct a sensitivity analysis on the truthful response rate. The results are reported in Figure S.3. We observe that both the MAE and the average interval length steadily decrease as $r$ increases, while the ECP remains close to or above the nominal 95% level across a wide range of response rates.

Second, we investigate the effect of the step-size schedule. We adopt

$$\gamma_m = \frac{20\,\bar{r}}{m^a + 100},$$

where $\bar{r}$ denotes the average truthful response rate and $a > 0$ controls the decay speed. Figure S.4 summarizes the results. Under the same experimental setting, the MAE, ECP, and Avg Len (average confidence-interval lengths) remain comparable across different values of $a$. This indicates that the proposed estimator is stable with respect to the choice of the step-size schedule.

## C.3 CONSERVATIVENESS OF THE SELF-NORMALIZED INFERENCE PROCEDURE

To assess the efficiency and conservativeness of the resulting confidence intervals, we compare the resulting confidence intervals with oracle normal-based intervals that use the true asymptotic variance derived in Theorem 3.1. Tables S.8–S.9 report the empirical coverage probabilities (ECP) and average interval lengths (Avg Len) for both types of intervals under normal and Cauchy data-generating distributions. We observe that compared with the oracle intervals, the self-normalized

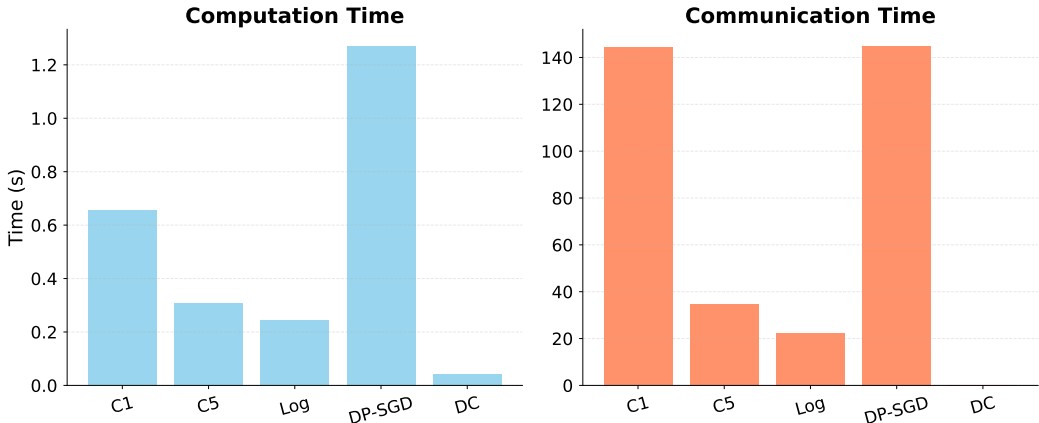

Figure S.2: Computation time and communication time for different methods. We fix $K = 10$ and $t_T = 10{,}000$.

intervals are slightly conservative: they achieve coverage at or above the nominal 95% level but produce moderately wider intervals. This mild conservativeness arises because self-normalization induces heavier-tailed limiting distributions compared with the standard normal approximation. It is important to note, however, that estimating the oracle variance in practice typically requires additional procedures that may consume further privacy budget under LDP. In contrast, the self-normalized construction avoids any variance estimation and remains fully privacy-preserving.

### C.4 EXTENSION TO DECENTRALIZED FEDERATED LEARNING

In this subsection, we provide an initial exploration of extending our method to a fully decentralized federated learning setting, where no central server is available and each client communicates only with its neighbors. Let the $K$ clients be connected through an undirected communication graph represented by an adjacency matrix $A = (a_{k_1 k_2}) \in \mathbb{R}^{K \times K}$, where $a_{k_1 k_2} = 1$ if client $k_1$ can exchange messages with client $k_2$ and $a_{k_1 k_2} = 0$ otherwise. Here we assume that every client is connected to itself, i.e., $a_{kk} = 1$, $k = 1, \ldots, K$. Define the degree $d_{k_1} = \sum_{k_2} a_{k_1 k_2}$ and construct the row-stochastic weight matrix $W = (w_{k_1 k_2})$ as

$$w_{k_1 k_2} = \frac{a_{k_1 k_2}}{d_{k_1}},$$

which encodes the network topology and performs neighbor averaging.

During communication iterations $t \in \mathcal{I}$, the aggregation step in Algorithm 1 is replaced by the following *weighted decentralized averaging*:

$$q_{t_m}^k \leftarrow \sum_{j=1}^{K} w_{kj} \, p_j \, q_{t_m}^j, \qquad k = 1, \ldots, K.$$

The local SGD update for $t \notin \mathcal{I}$ remains unchanged, yielding a fully decentralized variant of our estimator.

As a preliminary empirical study of this extension, we evaluate its finite-sample performance under a ring network topology, where each client communicates only with itself and its two immediate neighbors. All other experimental settings follow Section 4.1. The results in Table S.10 show that the decentralized estimator exhibits qualitatively similar behavior to its federated counterpart, suggesting that our method can naturally extend to decentralized learning. A complete theoretical analysis of this decentralized variant is left for future work.

| Scenario | Quantile ($\tau$) | Rate ($r$) | C1 | C5 | Log |
|---|---|---|---|---|---|
| | | | $t_T = 10\,000$ | | |
| | 0.5 | 0.25 | 0.950(0.0186) | 0.971(0.0292) | 0.979(0.0417) |
| | 0.5 | hetero | 0.942(0.0102) | 0.976(0.0140) | 0.993(0.0194) |
| | 0.5 | 0.9 | 0.955(0.0047) | 0.988(0.0062) | 0.998(0.0087) |
| | $\tau_{low}$ | 0.25 | 0.949(0.0190) | 0.975(0.0305) | 0.984(0.0433) |
| Homo | $\tau_{low}$ | hetero | 0.953(0.0097) | 0.981(0.0146) | 0.994(0.0203) |
| | $\tau_{low}$ | 0.9 | 0.985(0.0032) | 0.999(0.0064) | 1.000(0.0091) |
| | $\tau_{high}$ | 0.25 | 0.947(0.0208) | 0.971(0.0316) | 0.982(0.0461) |
| | $\tau_{high}$ | hetero | 0.928(0.0126) | 0.984(0.0155) | 0.993(0.0219) |
| | $\tau_{high}$ | 0.9 | 0.973(0.0058) | 0.997(0.0062) | 1.000(0.0100) |
| | 0.3 | 0.25 | 0.942(0.0255) | 0.971(0.0401) | 0.972(0.0565) |
| | 0.3 | hetero | 0.951(0.0127) | 0.968(0.0198) | 0.983(0.0258) |
| | 0.3 | 0.9 | 0.946(0.0054) | 0.961(0.0091) | 0.990(0.0119) |
| | 0.5 | 0.25 | 0.955(0.0225) | 0.970(0.0375) | 0.984(0.0531) |
| Hete L | 0.5 | hetero | 0.946(0.0116) | 0.981(0.0180) | 0.997(0.0247) |
| | 0.5 | 0.9 | 0.978(0.0044) | 0.993(0.0078) | 0.995(0.0119) |
| | 0.8 | 0.25 | 0.955(0.0339) | 0.975(0.0614) | 0.986(0.0890) |
| | 0.8 | hetero | 0.964(0.0171) | 0.987(0.0279) | 0.992(0.0387) |
| | 0.8 | 0.9 | 0.998(0.0050) | 0.988(0.0147) | 0.993(0.0197) |
| | | | $t_T = 50\,000$ | | |
| | 0.5 | 0.25 | 0.962(0.0081) | 0.984(0.0106) | 0.998(0.0158) |
| | 0.5 | hetero | 0.962(0.0040) | 0.988(0.0055) | 0.996(0.0080) |
| | 0.5 | 0.9 | 1.000(0.0012) | 1.000(0.0017) | 1.000(0.0028) |
| | $\tau_{low}$ | 0.25 | 0.945(0.0090) | 0.986(0.0110) | 0.994(0.0163) |
| Homo | $\tau_{low}$ | hetero | 0.970(0.0047) | 0.989(0.0058) | 0.995(0.0086) |
| | $\tau_{low}$ | 0.9 | 1.000(0.0020) | 1.000(0.0020) | 1.000(0.0030) |
| | $\tau_{high}$ | 0.25 | 0.950(0.0088) | 0.978(0.0118) | 0.998(0.0176) |
| | $\tau_{high}$ | hetero | 0.942(0.0044) | 0.983(0.0062) | 0.997(0.0090) |
| | $\tau_{high}$ | 0.9 | 0.984(0.0017) | 0.998(0.0036) | 0.999(0.0056) |
| | 0.3 | 0.25 | 0.943(0.0123) | 0.974(0.0155) | 0.990(0.0228) |
| | 0.3 | hetero | 0.937(0.0067) | 0.982(0.0071) | 0.994(0.0106) |
| | 0.3 | 0.9 | 0.992(0.0022) | 0.991(0.0029) | 0.997(0.0045) |
| | 0.5 | 0.25 | 0.952(0.0099) | 0.981(0.0136) | 0.994(0.0202) |
| Hete L | 0.5 | hetero | 0.966(0.0051) | 0.974(0.0065) | 0.992(0.0093) |
| | 0.5 | 0.9 | 0.978(0.0021) | 0.974(0.0028) | 0.993(0.0040) |
| | 0.8 | 0.25 | 0.948(0.0152) | 0.984(0.0207) | 0.990(0.0300) |
| | 0.8 | hetero | 0.949(0.0074) | 0.962(0.0102) | 0.995(0.0139) |
| | 0.8 | 0.9 | 0.902(0.0035) | 0.954(0.0084) | 0.991(0.0097) |

Table S.6: Empirical coverage probabilities at the 95% nominal level (mean absolute errors ↓) under varying quantile levels and truthful response rates, with different values of $t_T$ and a fixed number of $K = 10$ clients. Only 5 out of the 10 clients are randomly selected to participate in each communication round. In Homo, data for each client $k$ are independently drawn from $\mathcal{N}(0, 1)$. In Hete L, client-specific data are independently drawn from $\mathcal{N}(\mu_k, 1)$ with $\mu_k \sim \mathcal{N}(0, 1)$. The label "hetero" indicates heterogeneous truthful response rates with $r_k$ ranging uniformly from 0.25 to 0.9.

| Scenario | Quantile ($\tau$) | Rate ($r$) | C1 | C5 | Log |
|---|---|---|---|---|---|
| | | | $t_T = 10\,000$ | | |
| | 0.5 | 0.25 | 0.950(0.0232) | 0.973(0.0353) | 0.968(0.0473) |
| | 0.5 | hetero | 0.966(0.0114) | 0.991(0.0169) | 0.991(0.0229) |
| | 0.5 | 0.9 | 1.000(0.0034) | 1.000(0.0062) | 0.999(0.0100) |
| | $\tau_{\text{low}}$ | 0.25 | 0.936(0.0279) | 0.974(0.0393) | 0.967(0.0509) |
| Homo | $\tau_{\text{low}}$ | hetero | 0.928(0.0150) | 0.989(0.0187) | 0.988(0.0241) |
| | $\tau_{\text{low}}$ | 0.9 | 0.997(0.0048) | 1.000(0.0068) | 1.000(0.0105) |
| | $\tau_{\text{high}}$ | 0.25 | 0.940(0.0333) | 0.965(0.0466) | 0.961(0.0621) |
| | $\tau_{\text{high}}$ | hetero | 0.951(0.0184) | 0.985(0.0246) | 0.980(0.0338) |
| | $\tau_{\text{high}}$ | 0.9 | 1.000(0.0045) | 1.000(0.0088) | 0.999(0.0132) |
| | 0.3 | 0.25 | 0.947(0.0396) | 0.968(0.0588) | 0.952(0.0763) |
| | 0.3 | hetero | 0.956(0.0199) | 0.984(0.0306) | 0.993(0.0397) |
| | 0.3 | 0.9 | 0.999(0.0069) | 0.999(0.0257) | 1.000(0.0356) |
| | 0.5 | 0.25 | 0.940(0.0309) | 0.963(0.0432) | 0.968(0.0561) |
| Hete L | 0.5 | hetero | 0.965(0.0148) | 0.989(0.0220) | 0.980(0.0284) |
| | 0.5 | 0.9 | 0.999(0.0046) | 0.998(0.0076) | 0.998(0.0112) |
| | 0.8 | 0.25 | 0.935(0.0743) | 0.956(0.1039) | 0.956(0.1376) |
| | 0.8 | hetero | 0.947(0.0423) | 0.975(0.0539) | 0.972(0.0687) |
| | 0.8 | 0.9 | 0.993(0.0114) | 0.996(0.0145) | 0.990(0.0192) |
| | | | $t_T = 50\,000$ | | |
| | 0.5 | 0.25 | 0.943(0.0106) | 0.985(0.0148) | 0.984(0.0211) |
| | 0.5 | hetero | 0.970(0.0053) | 0.986(0.0087) | 0.989(0.0121) |
| | 0.5 | 0.9 | 1.000(0.0023) | 1.000(0.0050) | 0.997(0.0072) |
| | $\tau_{\text{low}}$ | 0.25 | 0.896(0.0147) | 0.988(0.0178) | 0.986(0.0248) |
| Homo | $\tau_{\text{low}}$ | hetero | 0.902(0.0087) | 0.980(0.0092) | 0.991(0.0124) |
| | $\tau_{\text{low}}$ | 0.9 | 1.000(0.0028) | 1.000(0.0027) | 0.999(0.0044) |
| | $\tau_{\text{high}}$ | 0.25 | 0.893(0.0205) | 0.982(0.0293) | 0.971(0.0431) |
| | $\tau_{\text{high}}$ | hetero | 0.930(0.0130) | 0.990(0.0212) | 0.985(0.0316) |
| | $\tau_{\text{high}}$ | 0.9 | 1.000(0.0046) | 1.000(0.0104) | 1.000(0.0161) |
| | 0.3 | 0.25 | 0.895(0.0232) | 0.970(0.0272) | 0.989(0.0377) |
| | 0.3 | hetero | 0.875(0.0138) | 0.983(0.0140) | 0.987(0.0188) |
| | 0.3 | 0.9 | 1.000(0.0046) | 1.000(0.0047) | 0.990(0.0089) |
| | 0.5 | 0.25 | 0.957(0.0136) | 0.984(0.0193) | 0.985(0.0274) |
| Hete L | 0.5 | hetero | 0.988(0.0068) | 0.991(0.0100) | 0.991(0.0154) |
| | 0.5 | 0.9 | 1.000(0.0031) | 1.000(0.0045) | 0.990(0.0090) |
| | 0.8 | 0.25 | 0.923(0.0442) | 0.972(0.0659) | 0.943(0.0969) |
| | 0.8 | hetero | 0.932(0.0284) | 0.987(0.0444) | 0.966(0.0678) |
| | 0.8 | 0.9 | 1.000(0.0089) | 1.000(0.0182) | 0.999(0.0299) |

Table S.7: Empirical coverage probabilities at the 95% nominal level (mean absolute errors $\downarrow$) under varying quantile levels and truthful response rates, with different values of $t_T$ and a fixed number of $K = 10$ clients. Only 5 out of the 10 clients are randomly selected to participate in each communication round. In Homo, data for each client $k$ are independently drawn from $\mathcal{C}(0, 1)$. In Hete L, client-specific data are independently drawn from $\mathcal{C}(\mu_k, 1)$ with $\mu_k \sim \mathcal{N}(0, 1)$. The label "hetero" indicates heterogeneous truthful response rates with $r_k$ ranging uniformly from 0.25 to 0.9.

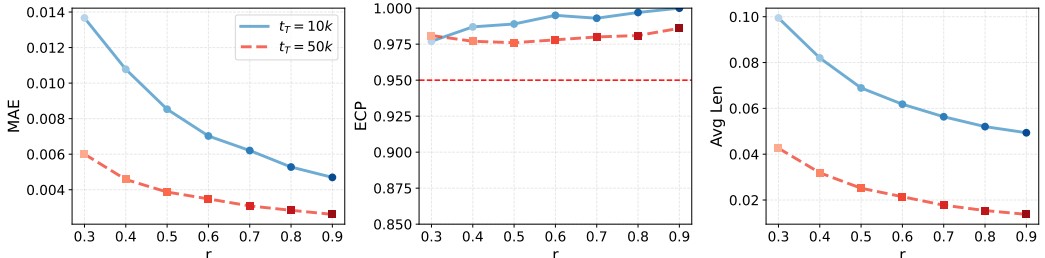

Figure S.3: Mean absolute errors (MAE ↓), empirical coverage probabilities at the 95% nominal level (ECP), and averaged confidence interval lengths (Avg Len ↓) under varying response rates. We fix $\tau_k \equiv \tau = 0.5$ and $K = 10$, and report results for different values of $t_T$ with data generated from $\mathcal{N}(0, 1)$.

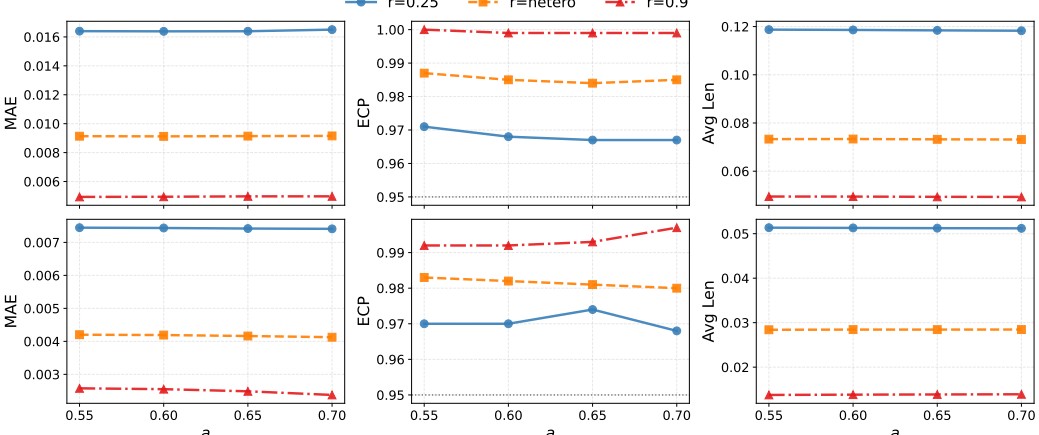

Figure S.4: Mean absolute errors (MAE ↓), empirical coverage probabilities at the 95% nominal level (ECP), and averaged confidence interval lengths (Avg Len ↓) under varying step-size parameter $a$. We fix $\tau_k \equiv \tau = 0.5$ and $K = 10$, with data generated from $\mathcal{N}(0, 1)$. Three response-rate scenarios are considered, where "hetero" indicates heterogeneous client-specific truthful response rates $r_k$ ranging uniformly from 0.25 to 0.9. The upper panels correspond to $t_T = 10,000$, and the lower panels to $t_T = 50,000$.

| Scenario | Quantile ($\tau$) | Rate ($r$) | Normal | | SN | |
|---|---|---|---|---|---|---|
| | | | ECP | Avg Len $(10^{-2})$ | ECP | Avg Len $(10^{-2})$ |
| | 0.5 | 0.25 | 0.930 | 6.215 | 0.949 | 8.792 |
| | 0.5 | hetero | 0.946 | 3.452 | 0.963 | 4.566 |
| | 0.5 | 0.9 | 0.996 | 1.726 | 0.995 | 1.909 |
| | $\tau_{\text{low}}$ | 0.25 | 0.928 | 6.411 | 0.947 | 9.135 |
| Homo | $\tau_{\text{low}}$ | hetero | 0.947 | 3.550 | 0.962 | 4.709 |
| | $\tau_{\text{low}}$ | 0.9 | 0.999 | 1.753 | 0.999 | 2.093 |
| | $\tau_{\text{high}}$ | 0.25 | 0.944 | 6.681 | 0.939 | 9.535 |
| | $\tau_{\text{high}}$ | hetero | 0.942 | 3.686 | 0.968 | 4.914 |
| | $\tau_{\text{high}}$ | 0.9 | 0.996 | 1.791 | 0.996 | 2.232 |
| | 0.3 | 0.25 | 0.930 | 8.504 | 0.958 | 12.090 |
| | 0.3 | hetero | 0.951 | 4.613 | 0.949 | 6.404 |
| | 0.3 | 0.9 | 0.996 | 2.053 | 1.000 | 3.228 |
| | 0.5 | 0.25 | 0.972 | 9.439 | 0.950 | 10.891 |
| Hete L | 0.5 | hetero | 0.987 | 5.099 | 0.952 | 5.604 |
| | 0.5 | 0.9 | 1.000 | 2.200 | 0.996 | 2.429 |
| | 0.8 | 0.25 | 0.997 | 19.021 | 0.966 | 17.014 |
| | 0.8 | hetero | 1.000 | 10.029 | 0.962 | 8.792 |
| | 0.8 | 0.9 | 0.999 | 3.488 | 0.990 | 4.759 |
| | 0.5 | 0.25 | 0.938 | 6.246 | 0.949 | 8.848 |
| Hete D | 0.5 | hetero | 0.958 | 3.469 | 0.966 | 4.788 |
| | 0.5 | 0.9 | 0.998 | 1.735 | 1.000 | 2.524 |

Table S.8: Empirical coverage probabilities at the 95% nominal level (ECP) and average confidence-interval lengths (Avg Len $\downarrow$) for the self-normalized and oracle normal-based confidence intervals. We fix $K = 10$, $E'_m = 1$, and $t_T = 10{,}000$. Three heterogeneity scenarios are considered. In Homo, data for each client $k$ are independently generated from $\mathcal{N}(0, 1)$. In Hete L, data for each client $k$ are independently generated from $\mathcal{N}(\mu_k, 1)$, where $\mu_k \sim \mathcal{N}(0, 1)$. In Hete D, data are generated from $\mathcal{N}(0, 1)$, $\mathcal{U}(-1, 1)$, and $\mathcal{C}(0, 1)$ across different clients. "hetero" indicates client-specific truthful response rates $r_k$ range uniformly from $[0.25, 0.9]$.

| Scenario | Quantile ($\tau$) | Rate ($r$) | Normal | | SN | |
|---|---|---|---|---|---|---|
| | | | ECP | Avg Len $(10^{-2})$ | ECP | Avg Len $(10^{-2})$ |
| Homo | 0.5 | 0.25 | 0.940 | 7.792 | 0.950 | 10.398 |
| | 0.5 | hetero | 0.953 | 4.326 | 0.953 | 5.906 |
| | 0.5 | 0.9 | 0.999 | 2.164 | 1.000 | 3.290 |
| | $\tau_{\text{low}}$ | 0.25 | 0.921 | 8.657 | 0.929 | 11.436 |
| | $\tau_{\text{low}}$ | hetero | 0.897 | 4.788 | 0.923 | 6.215 |
| | $\tau_{\text{low}}$ | 0.9 | 0.983 | 2.360 | 0.969 | 3.222 |
| | $\tau_{\text{high}}$ | 0.25 | 0.946 | 9.811 | 0.956 | 13.120 |
| | $\tau_{\text{high}}$ | hetero | 0.955 | 5.400 | 0.956 | 7.578 |
| | $\tau_{\text{high}}$ | 0.9 | 0.995 | 2.614 | 0.999 | 4.119 |
| Hete L | 0.3 | 0.25 | 0.923 | 12.044 | 0.947 | 17.008 |
| | 0.3 | hetero | 0.947 | 6.600 | 0.957 | 9.670 |
| | 0.3 | 0.9 | 0.999 | 3.105 | 1.000 | 5.241 |
| | 0.5 | 0.25 | 0.975 | 12.495 | 0.942 | 13.759 |
| | 0.5 | hetero | 0.971 | 6.835 | 0.946 | 8.276 |
| | 0.5 | 0.9 | 0.952 | 3.179 | 0.976 | 5.085 |
| | 0.8 | 0.25 | 0.999 | 38.893 | 0.935 | 28.048 |
| | 0.8 | hetero | 1.000 | 20.716 | 0.956 | 17.456 |
| | 0.8 | 0.9 | 0.998 | 8.240 | 0.985 | 10.558 |

Table S.9: Empirical coverage probabilities at the 95% (ECP) and average confidence-interval lengths (Avg Len ↓) for the self-normalized and oracle normal-based confidence intervals. We fix $K = 10$, $E'_m = 1$, and $t_T = 10{,}000$. In Homo, data for each client $k$ are independently generated from $\mathcal{C}(0,1)$. In Hete L, data for each client $k$ are independently generated from $\mathcal{C}(\mu_k, 1)$, where $\mu_k \sim \mathcal{N}(0,1)$. "hetero" indicates client-specific truthful response rates $r_k$ range uniformly from $[0.25, 0.9]$.

| Scenario | Quantile ($\tau$) | Rate (r) | C1 | C5 | Log |
|---|---|---|---|---|---|
| | | | $t_T = 10000$ | | |
| Hete L | 0.3 | hetero | 0.937(0.0178) | 0.969(0.0271) | 0.970(0.0352) |
| | 0.5 | hetero | 0.920(0.0142) | 0.964(0.0206) | 0.974(0.0260) |
| | 0.8 | hetero | 0.943(0.0261) | 0.975(0.0395) | 0.975(0.0515) |
| Hete D | 0.5 | 0.25 | 0.941(0.0141) | 0.977(0.0246) | 0.988(0.0362) |
| | 0.5 | hetero | 0.956(0.0073) | 0.981(0.0124) | 0.992(0.0172) |
| | 0.5 | 0.90 | 0.942(0.0039) | 0.994(0.0067) | 0.999(0.0097) |
| | | | $t_T = 50000$ | | |
| Hete L | 0.3 | hetero | 0.918(0.0093) | 0.950(0.0131) | 0.962(0.0178) |
| | 0.5 | hetero | 0.901(0.0073) | 0.959(0.0101) | 0.972(0.0136) |
| | 0.8 | hetero | 0.908(0.0142) | 0.968(0.0191) | 0.978(0.0270) |
| Hete D | 0.5 | 0.25 | 0.941(0.0061) | 0.977(0.0082) | 0.997(0.0127) |
| | 0.5 | hetero | 0.940(0.0034) | 0.977(0.0043) | 0.997(0.0063) |
| | 0.5 | 0.90 | 0.958(0.0016) | 0.978(0.0023) | 0.998(0.0035) |

Table S.10: Empirical coverage probabilities at the 95% nominal level (mean absolute errors ↓) under varying quantile levels and truthful response rates, evaluated under a decentralized ring topology with different values of $t_T$ and a fixed number of $K = 10$ clients. In Hete L, client-specific data are independently drawn from $\mathcal{C}(\mu_k, 1)$ with $\mu_k \sim \mathcal{N}(0,1)$. In Hete D, data are generated from $\mathcal{N}(0,1), \mathcal{U}(-1,1)$, and $\mathcal{C}(0,1)$ across different clients. The label "hetero" indicates heterogeneous truthful response rates with $r_k$ ranging uniformly from 0.25 to 0.9.

