# OpenReview forum: "Federated Learning of Quantile Inference under Local Differential Privacy"
_ICLR.cc/2026/Conference — ICLR 2026 Poster_

### Official Review · Reviewer_2i31 · 2025-10-28

**Soundness:** 3
**Presentation:** 3
**Contribution:** 3
**Rating:** 6
**Confidence:** 3

**Summary:**

The authors consider the problem of estimating a quantile in a federated learning setting with a local differential privacy constraint.  They present a new algorithm for computing this central quantile and extensive asymptotic analysis that can be used for statistical inference.  They highlight how estimating the quantile in this LDP setting is equivalent to estimating a quantile in a non private setting, but with the distribution shifted in a known way (i.e. solving one gives a computable solution to the other).  To get around estimating nuisance parameters, the authors also present a self-normalized approach based on a functional central limit theorem.

**Strengths:**

Overall the paper tackles an interesting problem and presents extensive mathematical results.

**Weaknesses:**

The organization and writing of the paper is interesting, but feels terse or missing in details at times.  See questions.

**Questions:**

Why in Theorem 3.2 do you use $\ell^\infty$ instead of $D[0,1]$ (the Skorohod space)?  My understanding is that a partial sum process like this is not tight in $\ell^\infty$ so you have to work in $D[0,1]$.  Or if you want to work in $\ell^\infty$ you can linearly interpolate the partial sum process so that you have continuous paths (and then you are essentially working in C[0,1]).  But as stated, I'm not sure Thm 3.2 is actually correct since a lack of tightness implies you don't have a functional CLT (in the given space).  But let me know if I'm missing something as it's been a while since I worked on functional CLTs.

I would argue that (2.1) is not the goal, but that the goal is outlined in lines 147-149.  It just so happens that the minimizer of (2.1) gives you the desired parameter,  i.e. the goal is to estimate the quantile of some target population.

The discussion around Thm 2.1 is quite weak and should be expanded.

The discussion/motivation for Thm 3.2 could be strengthened before being presented.  Just saying "it is necessary to strengthen" it, isn't really motivating the result.

Line 48: "Guarantees convergence" is a bit vague.  Statistically or computationally?

Line 61: Nasr should be in parentheses.  In general, the paper's handling of references is really all over the place.  References out of the sentence should be in parentheses, those that are part of the sentence shouldn't be, and you shouldn't have double parentheses like …(e.g.(Zhao…  I suggest the authors go through and clean these up.

Ln 124: When talking about estimation accuracy, I think the privacy parameters should be in the rate as well.  The choice of $\epsilon$ or $\delta$ play a large practical role.

Ln 130: The author should note the $\theta$ here is the one computed from the data, as opposed to the population level quantity.

Ln 151-161: The setup for the estimator is not very clear.  I suppose the authors are saying that at each step each client either updates locally or uses the current estimates from all of the other clients. But it is presented in a confusing fashion.  E.g. "$x_k^t$ is an independent realization of P_k", yes, but it is also just one of the datapoints held by a client.  "$E_m = n$" is divide an conquer?  So all clients have the same sample sizes?  I suppose $\mathcal I$ is ordered.  I would just add more details for the reader.

---

> ### Author Response · Authors · 2025-11-21
>
> We sincerely appreciate the time and effort you've invested in reviewing our paper. Your feedback is invaluable to us, and we have addressed each of your concerns below. Following your advice, we have carefully revised the manuscript, with all modifications highlighted in blue. Should you have any further questions, please do not hesitate to ask.
>
>
> **Weaknesses.**
>
> - The organization and writing of the paper are interesting, but at times feel terse or lacking in detail. See questions.
>
>   **Response:** Thank you for your thoughtful comments. We have provided point-by-point responses to all of your questions and have revised the manuscript accordingly. We welcome any further questions or suggestions.
>
> **Questions.**
>
> - Q1. Why does Theorem 3.2 use $\ell^\infty$ instead of $D[0,1]$ (the Skorohod space)?   My understanding is that a partial sum process of this form is not tight in $\ell^\infty$, so typically one works in $D[0,1]$. If you wish to work in $\ell^\infty$, the partial sum process can be linearly interpolated to obtain continuous paths (placing the process in $C[0,1]$).  As written, I am not sure Theorem 3.2 is valid, since a lack of tightness prevents a functional CLT in that space.  Please let me know if I am missing something—it has been some time since I last worked on functional CLTs.
>
>
> **Response:**
>     We agree with you that “If you want to work in $\ell^\infty$, you can linearly interpolate the partial sum process so that you obtain continuous paths (which effectively puts you in $C[0,1]$).”
> As you mentioned, we linearly interpolate the partial-sum process to obtain a continuous path. We then apply invariance principles in the martingale central limit theorem to establish weak convergence, which in turn yields the tightness of the interpolated process. Finally, we transfer this tightness result back to the original process in $\ell^\infty[0,1]$.
>
> Recall that
> $$
> \varepsilon_m=\frac{1}{E_m}\sum_{t=t_m}^{t_{m+1}-1}\sum_{k=1}^{K}p_k(G_k(\bar q_{t_m},\zeta_t^k)-g(\bar q_{t_m})).
> $$
>
> Define
> $$
> U_T^2=\sum_{m=1}^{T}\mathbb E(\varepsilon_{m}^2|\mathcal F_{t_m})
> $$
> and
> $$
> \Xi_T(r):=U_T^{-1}\left(\sum_{m=1}^{i}\varepsilon_m +\frac{(rU_T^2-U_i^2)\varepsilon_{i+1}}{U_{i+1}^2-U_i^2}\right),\quad U_i^2 \le rU_T^2 < U_{i+1}^2.
> $$
>
> Then,
> $$
>     \frac{\sqrt{t_T}}{T}\sum_{m=1}^{h(r,T)}\varepsilon_m
>     =\frac{\sqrt{t_T}}{T} U_T
>     \Xi_T\left( \frac{U_{h(r,T)}^2}{U_T^2} \right).
> $$
>
> Following Lemma B.16 in [1], one obtains that
> $$
> \sup_{r\in[0,1]}
> \left|
> \frac{\sqrt{t_T}}{T} U_T
> \Xi_T\left( \frac{U_{h(r,T)}^2}{U_T^2} \right)
> -\frac{\sqrt{t_T}}{T} U_T \Xi_T(r)
> \right|
> \xrightarrow{\mathbb P}0.
> \tag{1}
> $$
>
> By evaluating the conditions in Lemma B.13 of [1], the invariance
> principles in the martingale CLT yield that
> $$
> \Xi_T(r)\xrightarrow{d}B(r) \quad \text{in } \mathcal C[0,1],
> $$
> which further gives
> $$
> \frac{\sqrt{t_T}}{T}U_T\Xi_T(r)\xrightarrow{d}
> \frac{
> \sqrt{\nu\sum_{k = 1}^Kp_k^2\left(r_k^{-2}-(2Q_k-1)^2\right)}
> }{
> 2 \sum_{k=1}^K p_k f_k(Q^{\star})
> }
> B(r).
> $$
>
> Thus, Theorem A.2 of [2] implies that  $ \\left\\{ (\sqrt{t_T}/T) U_T  \Xi_T(r) \\right\\}_{T \geq 1} $ is tight in $\mathcal{C}[0,1]$.
>
> Using this tightness property, we obtain the tightness of  $$\\left\\{(\sqrt{t_T}/T)U_T\Xi_T( U_{h(r,T)}^{2} / U_T^{2} )\\right\\}_{T\geq 1}$$
>
> following the arguments in the proof of Lemma B.16 of [1].
> Therefore, the proof of the functional CLT is completed.

---

> ### Author Response · Authors · 2025-11-21
>
> **Questions.**
>
> - Q2. I would argue that (2.1) is not the goal; rather, the goal is outlined in lines 147–149. It just so happens that the minimizer of (2.1) yields the desired parameter, i.e., the aim is to estimate the quantile of some target population.
>
>   **Response:** Thank you for your suggestion. We have reorganized the goal accordingly.
>
> - Q3. The discussion around Theorem 2.1 is quite weak and should be expanded.
>
>   **Response:** Thank you for your comments. We have added more explanation regarding Theorem 2.1.
>
> - Q4. The discussion and motivation for Theorem 3.2 could be strengthened before being presented. Simply stating that ``it is necessary to strengthen'' it is not really motivating the result.
>
>   **Response:**  Thank you for the suggestion. We have now strengthened the discussion and motivation for Theorem 3.2 as follows: ``To address this issue, we establish a functional central limit theorem in Theorem~3.2, which serves as the foundation for constructing pivotal statistics via self-normalization. This allows us to conduct inference without estimating nuisance parameters, thereby providing a direct motivation for strengthening the asymptotic result.''
>
> - Q5. Line 48: “Guarantees convergence’’ is a bit vague. Statistically or computationally?
>
>   **Response:** By “guarantees convergence,’’ we refer to statistical convergence: the theoretical results ensure that the estimator converges to the target quantity in the appropriate probabilistic sense.
>
> - Q6. Line 61: *Nasr* should be in parentheses. In general, the paper’s handling of references is inconsistent. References outside the sentence should be in parentheses; those that are syntactically part of the sentence should not be. Also, avoid double parentheses such as “...(e.g.\ ((Zhao...”. I suggest the authors clean these up.
>
>   **Response:** Thank you for your correction. We have revised and checked the citations.
>
> - Q7. Line 124:  When discussing estimation accuracy, the privacy parameters should appear in the rate as well. The choice of \( \varepsilon \) or \( \delta \) plays a large practical role.
>
>   **Response:** Thank you for the suggestion. We have added a discussion below Theorem 3.1 on how the privacy parameters affect estimation accuracy, and we also provide the convergence rate in which the privacy parameters explicitly appear.
>
> - Q8. Line 130: The authors should note that the $\theta$ here is the one computed from the data, as opposed to the population-level quantity.
>
>   **Response:** Thank you for the comments. We have changed the notation to $\theta_{\mathrm{nonDP}}$ to avoid confusion.
>
> - Q9. Lines 151–161: The setup for the estimator is not very clear. I suppose the authors are saying that at each step each client either updates locally or uses the current estimates from all other clients. But the exposition is confusing. For example, “$x_k^t$ is an independent realization of $P_k$”—yes, but it is also just one of the datapoints held by a client. “$E_m = n$” is divide-and-conquer? So all clients have the same sample sizes? I suppose $\mathcal{I}$ is ordered. I would recommend adding more details for the reader.
>
>   **Response:** Thank you for the helpful comments. We have revised the exposition to make the estimator setup clearer. For example, we now explicitly state that $x_t^k$ is an observation drawn from the local dataset held by client $k$. We also clarify that the communication index set $\mathcal{I} = \\{t_0, t_1, \dots, t_T\\}$ is an ordered sequence of iterations. In addition, we removed the ambiguous expression “$E_m = n$” and clarified that the divide-and-conquer setting corresponds to performing only one communication (i.e., $|\mathcal{I}| = 1 $); see lines 176–200 on Page 4. These revisions should eliminate the confusion and make the estimator description more transparent.
>
> **Reference**
>
> [1] Xiang Li, Jiadong Liang, Xiangyu Chang, and Zhihua Zhang. Statistical estimation and online
> inference via local sgd. In Conference on Learning Theory, pp. 1613–1661. PMLR, 2022.
>
> [2] Peter Hall and Christopher C Heyde. Martingale limit theory and its application. Academic press,
> 2014.

---

### Official Review · Reviewer_6bXx · 2025-10-31

**Soundness:** 3
**Presentation:** 3
**Contribution:** 2
**Rating:** 4
**Confidence:** 4

**Summary:**

The paper proposes a federated algorithm for quantile estimation and inference under
Local Differential Privacy (LDP) using local stochastic gradient descent (Local-SGD). The
method perturbs local gradients via a randomised-response mechanism that ensures client-
level privacy while maintaining statistical efficiency. Key theoretical contributions include:
(i) an equivalence theorem reducing the LDP problem to a non-private one with adjusted
quantile levels, (ii) asymptotic normality and functional central limit theorem (FCLT) for the
non-smooth quantile loss, and (iii) a self-normalised confidence interval method that avoids
estimating nuisance parameters. Experiments and a real-data application validate the
theoretical results.

**Strengths:**

• Novel formulation combining federated learning, quantile inference, and local differential
privacy — a previously unexplored combination.
• Theoretical depth: establishes asymptotic normality and FCLT for non-smooth quantile
loss without the average-smoothness assumption.
• Elegant reduction (Theorem 2.1) that transforms the privatised problem into an
equivalent non-private version, enabling classical analysis.
• Self-normalised inference avoids density estimation, conserving privacy budget and
improving practical applicability.
• Comprehensive experiments demonstrating good accuracy and confidence interval
coverage across various privacy levels and heterogeneity settings.
• Clear discussion of limitations and directions for future research.

**Weaknesses:**

• The theoretical results depend on several strong regularity assumptions, particularly
bounded densities and step-size control, which may limit practical generality.
• The self-normalized approach yields heavier-tailed distributions, producing conservative
confidence intervals and lower test power.
• Dependence on a central aggregation server restricts use in fully decentralized federated
systems.
• Empirical validation is limited to tabular, one-dimensional quantiles — no high-
dimensional or complex FL tasks are explored.

• Certain proof steps (e.g., adaptation of Li et al., 2022 assumptions) rely on claims without
complete derivation; these could be more explicit for reproducibility.

**Questions:**

• Provide a proof sketch for Theorem 2.1 explicitly showing how the randomized response
induces the r_k scaling and τ̃_k shift.
• Expand on the removal of Assumption 1 from Li et al. (2022) — clarify exactly which steps
are substituted by the new bound E|q̄−Q*|² ≲ γ_m.
• Include empirical analysis of conservative confidence intervals — e.g., actual vs nominal
coverage rates.
• Test robustness under heavy-tailed client data (e.g., Cauchy distributions) to validate
bounded-density assumptions.
• Discuss communication-computation trade-offs more quantitatively under different local
update frequencies E_m.

---

> ### Author Response · Authors · 2025-11-21
>
> We sincerely appreciate the time and effort you've invested in reviewing our paper. Your feedback is invaluable to us, and we have addressed each of your concerns below. Following your advice, we have carefully revised the manuscript, with all modifications highlighted in blue. Should you have any further questions, please do not hesitate to ask.
>
> **For Weaknesses**
>
> - **W1**. "The  .... generality."
>     - Regarding the assumption on the uniform boundedness  of the client density functions in Assumption 1, this requirement is in fact quite mild. Because each density function integrates to one, any client-specific density must be bounded on its support. Consequently, imposing a uniform upper bound across clients does not introduce additional practical restrictions. Moreover, all data-generating mechanisms considered in our simulation studies naturally satisfy this condition.
>     - Besides, Assumption 2 allows the effective step size $ \gamma_m $ to lie in a fairly large range; for example, one may take $ \gamma_m \asymp m^{-a} $ for any $ a \in (1/2,1) $. As  reported in our additional sensitivity studies; see Appendix C.2 and Figure S.4 for details, our results are robust to reasonable variations in this hyperparameter.
>     - Regarding Assumption 3, since the learning rate $ \eta_m $ usually  decays with $m$, typically at a polynomial rate, if $E_m$ is bounded with respect to $m$, then the effective step size $\gamma_m$ in Assumption 2 naturally becomes non-increasing.  Alternatively, if $E_m$ increases with $m$, as long as the growth is sufficiently slow, such as at a logarithmic rate as in our simulation studies, the effective step size $\gamma_m$ in Assumption 2 still remains non-increasing.
>
> - **W2**. "The  ... power."
>     - Thank you for your comments.  As noted in Section 5, self-normalization can lead to heavier-tailed limiting distributions compared to the standard normal.
>     - In addition, we now have added a comparison with the "oracle" normal-based intervals that use the true asymptotic variance from Theorem 3.1 to further highlight the limitation of self-normalization; see Appendix C.3 and Tables S.9 and S.10 for details. As expected,  compared with the oracle intervals, the self-normalized intervals produce moderately wider intervals. We kindly remark, however, that estimating the true asymptotic variance typically requires additional procedures that may consume further privacy budget under LDP. In contrast, the self-normalized construction avoids any variance estimation and remains fully privacy-preserving.
>     - Furthermore, we have expanded the discussion on potential approaches that may achieve asymptotic normality under LDP settings, drawing on the most recent developments, as detailed in Section 5.
>
> - **W3**. "Dependence ... systems."
>     - We have now added an initial exploration of extending our method to a fully decentralized setting, where each client communicates only with its neighbors. Specifically, we consider an undirected graph with adjacency matrix $A=(a_{k_1k_2})$, where $a_{k_1k_2}=1$ if clients $k_1$ and $k_2$ can communicate. Define $W=(w_{kj})$ with
> $
> w_{kj} = a_{kj}/\sum_{j} a_{kj}.
> $
> During communication iterations $t\in\mathcal{I}$, the aggregation step in Algorithm 1 is replaced by
> $$
> q_{t_m}^k \leftarrow \sum_{j=1}^K w_{kj}\, p_j\, q_{t_m}^j,
> \qquad k=1,\dots,K,
> $$
> while the local SGD update for $t\notin\mathcal{I}$ remains unchanged.
>     - We also evaluate this decentralized variant under a ring topology. The results (Table S.11) show behavior qualitatively similar to the federated counterpart; see Appendix C.4 for details.
>
> - **W4**. "Empirical ... explored."
>     - First, we clarify your question. If "high-dimensional quantiles" refers to multivariate random variables, then there are very few viable approaches under LDP, since a multivariate quantile is not a single scalar quantity. If it instead refers to estimating multiple quantile levels, a direct approach would be to request data at each level and apply a multivariate extension of the local SGD procedure. However, such a naive method would consume an excessive privacy budget because the quantile levels are inherently correlated, and the SGD-based approach cannot avoid the quantile crossing issue. Therefore, multilevel or high-dimensional quantile estimation lies beyond the scope of this manuscript, and remains an active direction we are exploring.
>     - For more complex federated learning scenarios, we now have added new experiments that evaluate the proposed method under a more realistic setting with partial client participation. Specifically, at each communication round, only 5 out of the 10 clients are randomly selected to participate. We are pleased to report that the results are highly encouraging. In addition, to further strengthen our simulation study, we have conducted a set of additional experiments, including visual results; see Appendix C for details.
>
> - **W5**. Please refer to Q1 and Q2.

---

> ### Author Response · Authors · 2025-11-21
>
> **For Questions**
>
> - **Q1**. "Provide a ... quantile $\tilde{\tau}_k.$"
>
>     **Response:** Thank you for the suggestion. We now provide a proof sketch for Theorem 2.1; see the proof of Theorem 2.1 in Appendix B for details.
>
> - **Q2**. "Expand on ..."
>
>     **Response:** By closely examining the proof of Li et al. (2022) [1], we find that their Assumption 3.1 is used exclusively to establish the $\mathcal L^2$ convergence rate and the almost sure convergence result in their Lemma B.2. The appearance of Assumption~1 in the conditions of their Lemma B.3, which derivs the functional weak convergence, is solely for enabling the application of Lemma B.2. In our analysis, however, the key bound $ \mathbb{E}\lvert \bar{q}_{t_m} - Q^\star \rvert^2 \lesssim \gamma_m $ is proved directly under our  equation (B.2). This bound fully replaces the role played by Lemma B.2 in  Li et al. (2022). Consequently, Assumption 3.1 of Li et al. (2022) is not needed in our theoretical development beyond this step, and the functional central limit theorem follows without invoking that assumption.
>
> - **Q3**. "Include empirical ... rates."
>
>     **Response:** We now have compared the self-normalized confidence intervals with oracle normal-based intervals that use the true asymptotic variance derived in Theorem 3.1; see Appendix C.3 and Tables S.9 and S.10 for details. We observe that compared with the oracle intervals, the self-normalized intervals are slightly conservative: they achieve coverage at or above the nominal 95\% level but produce moderately wider intervals.  We kindly remark, however, that estimating the oracle variance in practice typically requires additional procedures that may consume further privacy budget under LDP. In contrast, the self-normalized construction avoids any variance estimation and remains fully privacy-preserving.
>
> - **Q4**. "Test robustness ... assumptions."
>
>     **Response:**
>     - It is worth clarifying that the bounded-density assumption does not require the random variables themselves to have bounded support. Instead, it requires their probability density functions to be uniformly bounded. This condition is in fact very mild: any random variable that admits a density must satisfy that the density integrates to one over $\mathbb{R}$. Consequently, the density function of each individual client must be bounded on its support. Therefore, imposing a uniform upper bound across clients does not introduce additional practical restrictions.
>     - Furthermore, following your kind advice, we have added new experiments to test the robustness of our method under heavy-tailed client data. Specifically, we consider exactly the same setting as in Section 4.1, except that the local data are now generated from a Cauchy distribution. The corresponding results are reported in Appendix C.1, Tables S.3 and S.4. Qualitatively, the findings closely mirror those under the Gaussian setting across all cases. These results further demonstrate that our procedure is robust even when data are extremely heavy-tailed.
>
> - **Q5**. "Discuss  ... $E_m$."
>
>     **Response:** Thank you for this important suggestion. Following your advice, we have now added a detailed quantitative analysis of both computation and communication costs for the proposed estimators and the baseline methods. In each simulation, we record the total wall-clock time (computation + communication) and report the averages over 1,000 repetitions; see Appendix C.1 and Figure S.2 for details. The results show that, first, for the proposed methods, as the communication frequency increases, the MAE decreases, but both computation time and communication cost naturally grow, demonstrating a clear efficiency–accuracy trade-off. In addition, comparing across different methods, the proposed method attains a communication cost comparable to DP-SGD while requiring noticeably less computation time. The DC method incurs the lowest overall cost because it performs only a single aggregation step; however, it also exhibits the largest MAE and substantial under-coverage in several heterogeneous settings, as reported in Section 4.
>
> **Reference**
>
> [1] Li, Liang, Chang, Zhang. Statistical Estimation and Online Inference via Local SGD. COLT, 2022.

---

### Official Review · Reviewer_t17M · 2025-11-01

**Soundness:** 3
**Presentation:** 3
**Contribution:** 2
**Rating:** 4
**Confidence:** 2

**Summary:**

This paper addresses a significant and challenging problem at the intersection of federated learning, statistical inference, and privacy preservation: performing quantile inference under local differential privacy (LDP). The authors propose a novel federated learning algorithm based on local Stochastic Gradient Descent (SGD), where client gradients are perturbed using an LDP mechanism. The key innovation is a theoretical analysis that establishes the asymptotic normality and a functional central limit theorem (FCLT) for their estimator, even though the quantile loss function is non-smooth. Leveraging this FCLT, they develop a self-normalized inference procedure that constructs valid confidence intervals without needing to estimate the asymptotic variance directly, thus avoiding additional privacy costs. Extensive simulations and a real-data application on government salary data demonstrate the method's effectiveness under various heterogeneity settings.

**Strengths:**

1.  Important and Timely Problem: The combination of federated quantile estimation with the strong privacy guarantee of LDP is highly relevant. Quantile inference is crucial for understanding data heterogeneity (e.g., in healthcare or economics), and LDP is the gold standard for privacy in distributed settings where a central server is not trusted. Tackling inference under these constraints is a substantial contribution.
2.  Significant Theoretical Contribution: The paper's primary strength is its theoretical analysis. Proving a functional central limit theorem for a local SGD-based estimator with a non-smooth loss function is a non-trivial and valuable advancement. This provides a solid foundation for inference in a setting where standard smoothness assumptions fail.
3.  Practical and Elegant Inference Solution: The use of self-normalization to construct confidence intervals is a clever and highly practical solution. It bypasses the notoriously difficult problem of estimating the asymptotic variance under LDP, which would typically require additional privacy budget or complex techniques. The resulting method is computationally efficient and integrates seamlessly with the federated learning workflow.
4.  Comprehensive Experimental Evaluation: The paper validates its theoretical claims with extensive simulations covering a wide range of heterogeneity scenarios (in quantile levels, privacy budgets, and data distributions). The inclusion of a real-world dataset (government salaries) strengthens the practical relevance of the findings. Comparisons against appropriate baselines like DP-SGD and a divide-and-conquer method are well-executed.

**Weaknesses:**

1.  Reliance on Strong Assumptions: The theoretical analysis relies on assumptions such as the boundedness of the parameter space and the uniform boundedness of the client density functions. While common in theoretical proofs, the practical implications of these assumptions could be discussed further (e.g., how to ensure parameter space boundedness in practice).
2.  Inherent Conservatism of Self-Normalization: The authors correctly note that self-normalization can lead to heavier-tailed limiting distributions than the standard normal. This often results in conservative confidence intervals (wider than necessary), which can reduce statistical power in hypothesis testing. This is a well-known trade-off of the method but is an important limitation to highlight.
3.  Centralized Server Dependency: The proposed framework requires a central server for aggregation and synchronization. The algorithm's applicability to fully decentralized or peer-to-peer federated learning environments, an active area of research, is not explored.
4.  Limited Analysis of Communication/Computation Trade-offs: While different communication strategies (C1, C5, Log) are compared, a more detailed analysis of the total communication cost and wall-clock time (computation + communication) compared to baseline methods would be beneficial for assessing real-world efficiency.

**Questions:**

1.  Have you quantified the degree of conservatism introduced by the self-normalization approach in your experiments? Are there potential adjustments or alternative techniques to mitigate this while remaining within the LDP constraint?
2.  Can your theoretical framework be extended to more complex models, such as high-dimensional or conditional quantile regression? What would be the main challenges?
3.  How would the algorithm perform under more realistic, asynchronous client participation patterns (e.g., clients joining at random times)? Does the theory accommodate such scenarios?

---

> ### Author Response · Authors · 2025-11-21
>
> We sincerely appreciate the time and effort you've invested in reviewing our paper. Your feedback is invaluable to us, and we have addressed each of your concerns below. Following your advice, we have carefully revised the manuscript, with all modifications highlighted in blue. Should you have any further questions, please do not hesitate to ask.
>
> **For Weaknesses**
> 1. Reliance on Strong Assumptions
> -  **Response**: Thank you for the insightful comment.
>     - The boundedness of the parameter space is a standard technical condition  that facilitate the asymptotic analysis; see for instance [1]. In practice, parameter boundedness is routinely enforced in optimization procedures. A common approach is to restrict the feasible domain so that algorithms such as SGD operate within a predetermined region. For example, when estimating salary quantiles in our real-data application, it is natural to specify a plausible prior range for wage levels, and the optimization is then constrained to this range.
>     - Regarding the assumption on the uniform boundedness of the client density functions, this requirement is in fact quite mild. Since each density function integrates to one, every client specific density must be bounded on its support. Therefore, assuming a uniform upper bound across clients does not impose additional practical restrictions, and various data-generating mechanisms considered in our simulation studies comply with this condition.
>
> 2. Inherent Conservatism of Self-Normalization
> - **Response**: Thank you for your comments.
>     - As noted in Section 5, self-normalization can lead to heavier-tailed limiting distributions compared to the standard normal.
>     - In addition, we now have added a comparison with the "oracle" normal based intervals that use the true asymptotic variance from Theorem 3.1 to further highlight the limitation of self-normalization; see Tables S.9 and S.10 in Appendix C.3 for details.
>     - Furthermore, we have expanded the discussion on potential approaches that may achieve asymptotic normality under LDP settings, drawing on the most recent developments, as detailed in Section 5.
>
> 3. Centralized Server Dependency
> - **Response**: Thank you for the important comment.
>     - We have now added an initial exploration of extending our method to a fully decentralized federated learning setting, where no central server is available and each client communicates only with its neighbors. Specifically, we consider an undirected communication graph with adjacency matrix $A=(a_{k_1k_2})$, where $a_{k_1k_2}=1$ if clients $k_1$ and $k_2$ can communicate. Define $W=(w_{kj})$ with $w_{kj} = a_{kj}/\sum_{j} a_{kj}.$
> During communication iterations $t\in\mathcal{I}$, the aggregation step in Algorithm 1 is replaced by
> $$
> q_{t_m}^k \leftarrow \sum_{j=1}^K w_{kj} p_j q_{t_m}^j,
> \qquad k=1,\dots,K,
> $$ while the local SGD update for $t\notin\mathcal{I}$ remains unchanged.
>     - As a preliminary empirical study, we evaluate this decentralized variant under a ring topology, where each client communicates only with its two neighbors; all other settings follow Section 4.1. The results (Table S.11) show behavior qualitatively similar to the federated counterpart, suggesting that our method naturally extends to decentralized learning. A full theoretical investigation is left for future work; see Appendix C.4 for details.
>
> 4. Limited Analysis of Communication/Computation Trade-offs
> - **Response**: Thank you for this valuable suggestion. We have now added a detailed quantitative analysis of both computation and communication costs for the proposed estimators and the baseline methods. In each simulation, we record the total wall-clock time (computation + communication) and report the averages over 1,000 repetitions; see Appendix C.1 and Figure S.2 for details. The results clearly illustrate the expected efficiency–accuracy trade-off across communication strategies.

---

> > ### Author Response · Authors · 2025-11-21
> >
> > **For Questions**
> >
> > - **Q1.** Please refer to W1.
> >
> > - **Q2.**  Thank you for your questions.
> >     - The regression model is indeed interesting, and there are direct approaches such as DP-SGD in a single-server setting. For the low-dimensional case, one may apply this idea within local SGD together with our theory for non-smooth loss functions. A potential drawback is that the gradient-level privacy mechanism does not exploit the structure of the indicator function, which can lead to inefficiency.
> >     - For the high-dimensional case, beyond the increased privacy burden as the dimension grows, the main challenge lies in establishing CLT or Gaussian approximation results in high dimensions. To the best of our knowledge, the most recent relevant work is [2], which studies SGD in the single-server setting under stochastic Lipschitz continuity assumptions.
> >
> > - **Q3.** Thank you for raising this important question.
> >     - We have added new experiments to evaluate the proposed method under a more realistic setting with partial client participation. Specifically, at each communication round, only 5 out of the 10 clients are randomly selected to participate, while all other configurations remain identical to those in Section 4.1; see Appendix C.1 and Tables S.7 and S.8 for details. We observe that the empirical coverage probabilities remain close to the nominal 95\% level, and the MAE increases only mildly compared with the full-participation case. These findings indicate that the proposed estimator is robust to partial and asynchronous client participation.
> >     - From a theoretical perspective, our current results do not directly cover this scenario. Understanding how asynchronous client participation affects estimation and inference is an important direction for future work, and we have now explicitly highlighted this point in Section 5.
> >
> > **Reference**
> >
> > [1] Gu, J. and Chen, S. X. Distributed Statistical Inference under Heterogeneity. JMLR, 24(387), 2023.
> >
> > [2] Li, J., Lou, Z., Schmidt-Hieber, J., Wu, W. B. Statistical Guarantees for High-Dimensional SGD. NeurIPS, 2025.

---

### Official Review · Reviewer_eKT7 · 2025-11-02

**Soundness:** 2
**Presentation:** 3
**Contribution:** 2
**Rating:** 4
**Confidence:** 3

**Summary:**

This paper propose to use federated learning (FL) for quantile inference under local differential privacy (LDP). The core idea of this paper is to design an LDP mechanism that can transform the LDP federated quantile estimation into the non-DP case. In addition, via constructing valid confidence intervals under LDP, the authors develop a self-normalized inference procedure without requiring direct estimation of the asymptotic
variance. The extensive experiments evaluate the effectiveness of the proposed method. Overall, this paper considers a new problem in this area. However, I have some concerns as follows: 1) I cannot see any baselines in the experiments. Although the authors claim that this is the first inference framework for federated quantile estimation, related works are existing and also can be adopted as baselines; 2) The theoretical analysis is not complete, especially for DP. The authors claim that they transform the LDP federated quantile estimation into the non-DP case. Can this transformation maintain DP guarantee? 3) Existing results in simulation cannot evaluate the proposed methods well. Some more experimental should be added.

**Strengths:**

This paper propose to use federated learning (FL) for quantile inference under local differential privacy (LDP). The core idea of this paper is to design an LDP mechanism that can transform the LDP federated quantile estimation into the non-DP case. In addition, via constructing valid confidence intervals under LDP, the authors develop a self-normalized inference procedure without requiring direct estimation of the asymptotic
variance. The extensive experiments evaluate the effectiveness of the proposed method. Overall, this paper considers a new problem in this area and well written.

**Weaknesses:**

However, I have some concerns as follows: 1) I cannot see any baselines in the experiments. Although the authors claim that this is the first inference framework for federated quantile estimation, related works are existing and also can be adopted as baselines; 2) The theoretical analysis is not complete, especially for DP. The authors claim that they transform the LDP federated quantile estimation into the non-DP case. Can this transformation maintain DP guarantee? 3) Existing results in simulation cannot evaluate the proposed methods well. Some more experimental should be added.

**Questions:**

1) I cannot see any baselines in the experiments. Although the authors claim that this is the first inference framework for federated quantile estimation, related works are existing and also can be adopted as baselines;
2) The theoretical analysis is not complete, especially for DP. The authors claim that they transform the LDP federated quantile estimation into the non-DP case. Can this transformation maintain DP guarantee?
3) Existing results in simulation cannot evaluate the proposed methods well. Some more experimental should be added.

---

> ### Author Response · Authors · 2025-11-21
>
> We sincerely appreciate the time and effort you've invested in reviewing our paper. Your feedback is invaluable to us, and we have addressed each of your concerns below. Following your advice, we have carefully revised the manuscript, with all modifications highlighted in blue. Should you have any further questions, please do not hesitate to ask.
>
> **For Weaknesses**
>
> 1. "I do not  .... baselines."
>
> - **Response**: Thank you for your comment. Our experiments already include two baselines: (i) DP-SGD [1], which replaces randomized response by adding Laplace noise directly to the gradients; and (ii) Divide-and-Conquer (DC), corresponding to the extreme case with only one final communication round. Following your kind suggestion, we additionally include (iii) Single, the single-machine LDP quantile inference method of [2], where all data are held on a single device and LDP-SGD is performed without any federated communication; see the revised Table 1 on Page 9, as well as lines 419–431 on Page 8. We find the results highly encouraging, and they highlight the effectiveness of the proposed federated estimator.
>
> 2. "The theoretical  ... guarantee?"
>
> - **Response**: Thanks for your comments. We apologize for the potential confusion regarding Theorem 2.1. The main purpose of Theorem 2.1 is to provide theoretical insight showing that the LDP federated quantile estimation problem can be reduced to a non-DP federated quantile estimation problem through an appropriate transformation. Consequently, once one derives the local SGD method for the non-smooth loss, it can be directly applied to the LDP federated quantile estimation setting. Therefore, this step does not involve the LDP mechanism itself, and we still implement Algorithm 1. We have revised the corresponding paragraph.
>
>
> 3. "The existing ... included."
>
> - **Response**: Thank you for your comment. We have now substantially expanded our experimental evaluation to more comprehensively assess the proposed method. Specifically, we have added the following new simulation studies:
>
>     (i) additional Cauchy data-generating distributions to evaluate performance under heavy-tailed settings;
>
>     (ii) a comparison with an additional baseline: the single-machine LDP quantile inference method of [2];
>
>     (iii) a quantitative analysis of computation and communication efficiency to further evaluate the efficiency–accuracy trade-off;
>
>
>     (iv) we consider partial client participation to mimic more realistic federated environments;
>
>
>     (v) sensitivity analyses on the truthful response rate and step-size schedule to assess the robustness of the method;
>
>
>     (vi) a comparison between the resulting confidence intervals and oracle normal-based intervals to illustrate the conservativeness of the self-normalized inference procedure;
>
>
>     (vii) an initial exploration of extending our method to a fully decentralized federated learning setting; see Appendix~C for details. We are pleased to report that the results are highly encouraging. We believe these additional studies offer a much more comprehensive evaluation and further demonstrate the effectiveness and robustness of the proposed approach.
>
> **For Questions**
>
> - **Q1**. Please refer to W1
>
> - **Q2**. Please refer to W2
>
> - **Q3**. Please refer to W3
>
> **Reference**
>
> [1] Shuang Song, Kamalika Chaudhuri, and Anand D Sarwate. Stochastic gradient descent with differentially private updates. In 2013 IEEE global conference on signal and information processing, pp. 245–248. IEEE, 2013.
>
>
> [2] Yi Liu, Qirui Hu, Lei Ding, and Linglong Kong. Online local differential private quantile inference via self-normalization. In International Conference on Machine Learning, pp. 21698–21714. PMLR, 2023.

---

### Official Review · Reviewer_Betv · 2025-11-02

**Soundness:** 2
**Presentation:** 1
**Contribution:** 2
**Rating:** 4
**Confidence:** 3

**Summary:**

This paper considers the statistical inference of federated learning for quantile loss via local stochastic gradient decent under local differential privacy. It proposes a local-SGD algorithm with randomized response on the binary quantile “gradient” supports heterogeneous clients, and outputs a Polyak–Ruppert averaged estimator. Then, its theoretical properties were explored and experimentally verified.

**Strengths:**

S1. This work designs an LDP mechanism that can transform the LDP federated quantile estimation into the non-DP case, and then derive the asymptotic normality and functional central limit theorem of the proposed estimator under non-DP cases.

S2. It is the first weak-convergence result for local SGD without the usual average-smoothness assumption in existing literature

**Weaknesses:**

My comments focus on presentation and experimental design issues.

W1. The introduction section reads like an extended related work. Missing the importance and/or motivation of the question. I suggest giving 1–2 concrete use cases, or moving a simplified version of Figure 1 to the intro to anchor the problem.

W2. The Methodology section lacks early notation and problem formulation. Even Q is not defined.

W3. In the Experiment section, placing synthetic results in the appendix is a bad idea. There is no direct comparison against prior work (e.g., Liu et al, 2023b). Table 1 lacks guidance on “what good looks like”, nominal 95% coverage with the shortest average CI length? Why not visually highlight when empirical quantiles fall inside/outside CIs?

W4. For the truthful response rate, I am unsure who sets $r_k$, whether it is policy-driven, user-chosen, or purely synthetic. The phrase "range from 0.6 to 0.9" in real data is missing a clear definition. I suggest adding the sensitivity of the experiment regarding the truthful response rate.

W5. In Asymptotic Analysis, are the theorems and algorithms (partially) verified in the experimental section? If not, is verification possible?

**Questions:**

See W1-W5.

---

> ### Author Response · Authors · 2025-11-21
>
> We sincerely appreciate the time and effort you've invested in reviewing our paper. Your feedback is invaluable to us, and we have addressed each of your concerns below. Following your advice, we have carefully revised the manuscript, with all modifications highlighted in blue. Should you have any further questions, please do not hesitate to ask.
>
> **For Weaknesses**
>
> 1. "The introduction  .... problem."
>
>
> - **Response**: Thank you for your suggestions. We have revised the Introduction and corrected the errors to better highlight the motivation and importance, following your advice. The updated manuscript has been uploaded.
>
>
> 2. "The Methodology ... defined."
>
>
> - **Response**: Thank you for your comment. The variable $Q$ appears in the loss function. For better clarity, we have added a notation table in the Appendix and highlighted problem in introduction. Further, we adjust the problem formulation to better present.  The remaining clarity-related revisions are summarized in W1.
>
>
> 3. "In the Experiment ...  the CIs?"
>
>
>
> - **Response**: Thank you for the helpful comments, and we apologize for not making this point sufficiently clear.
>     - We have moved several key synthetic-data results into the main text to improve readability.
>     - Our experiments originally included two baselines: (i) DP-SGD [1], which replaces randomized response by adding Laplace noise directly to the gradients; and (ii) Divide-and-Conquer (DC), corresponding to the extreme case with only a single final communication round. Following your suggestion, we now additionally include (iii) Single, the single-machine LDP quantile inference method of [2].
>     - By “what good looks like,” our evaluation criteria are (i) empirical coverage probabilities (ECP) close to or above the nominal 95\% level, and (ii) mean absolute errors (MAE) as small as possible, indicating more accurate quantile estimation. We now state this explicitly in the table captions using the phrasing “Empirical coverage probabilities at the 95\% nominal level (mean absolute errors $\downarrow$) under $\dots$”.
>     - We also explicitly mark cases in which the empirical quantiles fall far below the nominal 95\% level, so that coverage failures can be directly identified; see, for example, the revised Table 1 on Page 9.
>
>
>
>
>
> 4. "For the truthful ... rate."
>
>
> - **Response**: We apologize for not making this point clear.
>     - In practice, the truthful response rate is policy-driven or user-chosen, and is therefore treated as a fixed quantity rather than a tuning parameter. In our experiments, different rates are used only to examine the robustness and usefulness of the proposed method under heterogeneous privacy conditions.
>     - The phrase "range from 0.6 to 0.9" refers to response rates $r_k \in {0.6,\text{hetero},0.9}$, where "hetero" indicates client-specific truthful response rates $r_k$ ranges uniformly from 0.6 to 0.9. This setting has been clarified in the revised manuscript.
>     - Following your suggestion, we have now added a sensitivity analysis on the truthful response rate under the same experimental setting in Section 4.1; see Appendix C.2 and Figure S.3 for details. We observe that the MAE and the average confidence-interval lengths steadily decrease as $r$ increases, while the ECP remains close to or above 95\% over a broad range of response rates.
>
>
> 5. "In the Asymptotic  ... possible?"
>
> - **Response**: In our experiments, we evaluated the self-normalized confidence intervals under a wide range of heterogeneity settings. The empirical coverage levels consistently exceeded the nominal $95\%$, which directly confirms Corollaries 3.1 and 3.2—both of which follow from our main Theorem 3.2. Since Theorem 3.2 is a functional CLT, directly verifying the weak convergence of the estimator process to a Brownian motion is infeasible in practice. Nevertheless, our experiments validate the correctness of its consequences obtained via the continuous mapping theorem. In addition, Theorem 3.1 is a special case of Theorem 3.2, as already noted in the manuscript.
>
>
> **For Questions**
>
> - **Response**: Please refer to W1 - W5.
>
> **Reference**
>
>
> [1] Shuang Song, Kamalika Chaudhuri, and Anand D Sarwate. Stochastic gradient descent with differentially private updates. In 2013 IEEE global conference on signal and information processing, pp. 245–248. IEEE, 2013.
>
>
> [2] Yi Liu, Qirui Hu, Lei Ding, and Linglong Kong. Online local differential private quantile inference via self-normalization. In International Conference on Machine Learning, pp. 21698–21714. PMLR, 2023.

---

### Author Response · Authors · 2025-12-02

We thank the reviewers for their time and insightful feedback on our paper, which, to the best of our knowledge, is the first to establish asymptotic confidence intervals for federated quantile estimation under LDP.

Across all reviewers, there is strong agreement that the paper tackles an **important and previously unexplored problem** at the intersection of federated learning, quantile inference, and local differential privacy:

- Reviewer t17M highlights the practical importance of combining quantile estimation with strong LDP guarantees, while Reviewer 6bXx notes that this FL–quantile–LDP formulation has not been studied before.

- Reviewers Betv, eKT7, and 6bXx all commend the novel LDP mechanism that transforms the privatized problem into an equivalent non-private version, enabling classical analysis.

**Theoretical contributions are widely acknowledged**:

- Reviewer Betv emphasizes the first weak convergence result for local SGD without average smoothness.

- Reviewer t17M highlights the challenge of proving an FCLT under a non-smooth loss.

- Reviewer 6bXx underscores the strength and depth of our asymptotic normality and FCLT results.

- Reviewers eKT7, t17M, and 6bXx also highlight the practical elegance of the self-normalized inference strategy, which avoids density estimation and preserves privacy budget.

**The experimental evaluation is uniformly praised**:

- Reviewer t17M highlights the broad heterogeneity scenarios and real data validation.

- Reviewer 6bXx commends the strong accuracy and confidence interval coverage.

- Reviewer eKT7 finds the empirical studies effective and the paper well written.

Collectively, the reviewers recognize the submission as a novel, theoretically rigorous, and practically impactful contribution to privacy preserving federated quantile inference.

---

The reviewers also raised valuable concerns regarding the clarity of presentation, explanations of theoretical results, and numerical evaluations. We have thoroughly addressed these in our rebuttal. **A summary of the revisions is as follows**.

**Presentation and discussion**:

We improved the theoretical explanations and intuitions, improved the presentation for better readability, and substantially revised the introduction to better motivate the problem and highlight our contributions.

**Numerical experiments**:

We have added a set of new simulation studies, including: (i) additional Cauchy data-generating distributions to evaluate performance under heavy-tailed settings; (ii) a comparison with an additional baseline; (iii) a quantitative analysis of computation and communication efficiency to further evaluate the efficiency–accuracy trade-off; (iv) partial client participation to mimic more realistic federated environments; (v) sensitivity analyses on the truthful response rate and step-size schedule to assess the robustness of the method; (vi) a comparison between the resulting confidence intervals and oracle normal-based intervals to illustrate the conservativeness of the self-normalized inference procedure; and (vii) an initial exploration of extending our method to a fully decentralized federated learning setting. We are pleased to report that the results are highly encouraging.

We have carefully revised the manuscript to incorporate the corresponding improvements, with all modifications highlighted in blue. We sincerely appreciate the constructive feedback and believe that these revisions significantly strengthen the paper.

---

### Meta-Review · Area_Chair_TP6v · 2026-01-08

**Summary:**

There is strong and broad agreement among the reviewers that this paper addresses a **novel and important problem**, namely, statistical inference for quantile estimation in federated learning under local differential privacy (LDP). Although the initial scores from Reviewers Betv, eKT7, t17M, and 6bXx were 4, **all major concerns were addressed** in detail during the rebuttal, often through additional comprehensive experiments, clarified proofs, or expanded discussion. If the reviewers had been able to participate fully in the discussion, I believe their evaluations would have moved toward an acceptance.

**Reviewer Concerns:**

**Reviewer Betv’s concerns:**

(1) The concerns regarding the presentation of the **Introduction** and **Methodology** were addressed by revising these sections and correcting errors, thereby better highlighting the motivation and significance of the work.

(2) The concerns regarding the **Experiment**, including Table 1 and the choice of $r_k$,

were addressed by moving several key synthetic data results into the main text, providing more detailed explanations, and adding a sensitivity analysis on the truthful response rate.

(3) The concern about **whether the theoretical results and algorithms are empirically (partially) verified** was partially addressed. The authors do not explicitly indicate which experiments correspond to or validate Corollaries 3.1 and 3.2, respectively.

**Reviewer eKT7’s concerns:**

(1) The concern regarding the **baselines** used in the experiments was addressed by providing more detailed explanations and by adding an additional baseline, referred to as Single.

(2) The concern regarding the **theoretical analysis** was addressed by including more detailed explanations following Theorem 2.1.

(3) The concern regarding the **experiments** was addressed by adding substantially more comprehensive experiments in Appendix C.

**Reviewer t17M’s concerns:**

(1) The concern regarding Reliance on Strong Assumptions was addressed through more detailed explanations, supported by real-data applications.

(2) The concern regarding Inherent Conservatism of Self-Normalization was addressed by additional discussion.

(3) The concern regarding Centralized Server Dependency was addressed by including an initial exploration of extending the proposed method to a fully decentralized federated learning setting in Appendix C.4.

(4) The concern regarding Limited Analysis of Communication/Computation Trade-offs was addressed by adding a detailed quantitative comparison of both computation and communication costs for the proposed estimators and the baseline methods in Appendix C.1.

(5) The concern regarding extension to more complex models was addressed through more detailed explanations and discussion.

(6) The concern regarding  more realistic, asynchronous client participation patterns was addressed by adding new experimental results in Appendix C.1.

(7) The remaining outstanding concern relates to Question 1, namely, Have the authors quantified the degree of conservatism introduced by the self-normalization approach in experiments? Are there potential adjustments or alternative techniques to mitigate this while remaining within the LDP constraint?

**Reviewer 6bXx’s concerns:**

(1) The concern regarding **assumptions** was addressed through more detailed explanations, supported by the simulation studies.

(2) The concern regarding **self-normalization** was addressed by adding further discussion and quantitative comparisons in Appendix C.3.

(3) The concern regarding **fully decentralized federated systems** was addressed by including an initial exploration of extending the proposed method to a fully decentralized federated learning setting in Appendix C.4.

(4) The concern regarding **high-dimensional or complex federated learning tasks** was addressed through more detailed explanations and by adding new experimental results in Appendix C.

(5) The concerns regarding **certain proof steps** were addressed by providing a proof sketch for Theorem 2.1 in Appendix B, along with additional explanatory details.

(6) The concern regarding **robustness under heavy-tailed client data** was addressed by adding new experimental results in Appendix C.1.

(7) The concern regarding **communication–computation trade-offs** was addressed by including a detailed quantitative comparison of both computation and communication costs for the proposed estimators and the baseline methods in Appendix C.1.

**Reviewer 2i31’s concerns:**

(1) The concern regarding the proof of Theorem 3.2 was addressed by providing a proof of the functional CLT, following Reviewer 2i31’s suggestion.

(2) The concerns regarding the goal of the paper, the discussion around Theorems 2.1 and 3.2, and citation issues were addressed through revisions and clarifications.

(3) The concern regarding the role of privacy parameters in estimation accuracy and convergence rate was addressed by adding a discussion following Theorem 3.1.

(4) The concern regarding the setup for the estimator was addressed by revising and clarifying the exposition in Section Methodology.

**Reviewer Scores:**

**Reviewer 2i31** would likely maintain a score of 6, as the concerns raised in the review were adequately addressed in the rebuttal.

**Reviewers Betv, eKT7, t17M, and 6bXx** would likely increase their scores (likely from 4 to 6), since the main concerns raised in their reviews were adequately addressed in the rebuttal, in particular through more detailed explanations and the inclusion of substantially more comprehensive experimental results in Appendix C.

---

### Decision · Program_Chairs · 2026-01-26

Accept (Poster)